# Resensitizing carbapenem- and colistin-resistant bacteria to antibiotics using auranofin

Hongzhe Sun [1,2,3,9✉], Qi Zhang[1,9], Runming Wang [1,9✉], Haibo Wang[1], Yuen-Ting Wong[1,4], Minji Wang [5], Quan Hao [6], Aixin Yan [5], Richard Yi-Tsun Kao[4,7,8], Pak-Leung Ho [4,7,8] & Hongyan Li[1]

Global emergence of Gram-negative bacteria carrying the plasmid-borne resistance genes, $bla_{MBL}$ and $mcr$, raises a significant challenge to the treatment of life-threatening infections by the antibiotics, carbapenem and colistin (COL). Here, we identify an antirheumatic drug, auranofin (AUR) as a dual inhibitor of metallo-β-lactamases (MBLs) and mobilized colistin resistance (MCRs), two resistance enzymes that have distinct structures and substrates. We demonstrate that AUR irreversibly abrogates both enzyme activity via the displacement of Zn (II) cofactors from their active sites. We further show that AUR synergizes with antibiotics on killing a broad spectrum of carbapenem and/or COL resistant bacterial strains, and slows down the development of β-lactam and COL resistance. Combination of AUR and COL rescues all mice infected by *Escherichia coli* co-expressing MCR-1 and New Delhi metallo-β-lactamase 5 (NDM-5). Our findings provide potential therapeutic strategy to combine AUR with antibiotics for combating superbugs co-producing MBLs and MCRs.

[1] Department of Chemistry, The University of Hong Kong, Pokfulam Road, Hong Kong, SAR, China. [2] State Key Laboratory of Synthetic Chemistry, The University of Hong Kong, Pokfulam Road, Hong Kong, SAR, China. [3] CAS-HKU Joint Laboratory of Metallomics on Health and Environment, The University of Hong Kong, The University of Hong Kong, Pokfulam Road, Hong Kong, SAR, China. [4] Department of Microbiology, The University of Hong Kong, Sassoon Road, Hong Kong, SAR, China. [5] School of Biological Sciences, The University of Hong Kong, Pokfulam Road, Hong Kong, SAR, China. [6] School of Biomedical Sciences, The University of Hong Kong, Sassoon Road, Hong Kong, SAR, China. [7] State Key Laboratory of Emerging Infectious Diseases, Carol Yu Centre for Infection, The University of Hong Kong, Hong Kong, SAR, China. [8] The Research Centre of Infection and Immunology, Li Ka Shing Faculty of Medicine, The University of Hong Kong, Hong Kong, SAR, China. [9]These authors contributed equally: Hongzhe Sun, Qi Zhang, Runming Wang. ✉email: hsun@hku.hk; u3002771@connect.hku.hk

The clinical efficacy of antibiotics has been severely challenged by plasmid-borne resistance determinants, in particular carbapenemase that renders Gram-negative bacteria resistant to carbapenem therapy, in either hospital or community settings, triggering the onset of the worldwide antimicrobial resistance crisis[1]. As one of the most clinically relevant carbapenemase, New Delhi metallo-β-lactamase 1 (NDM-1) has experienced the widest geographical spread since its discovery in 2008[2]. The resistance gene encoding NDM-1 ($bla_{NDM-1}$) could be rapidly disseminated via plasmid as well as integrons cassette[3] and expressed without apparent fitness cost among different bacteria[4]. NDM-1 as well as other MBLs, confers bacterial resistance to nearly all available β-lactam antibiotics, leading to poor clinical outcomes of conventional antibiotic therapy[5]. Moreover, most $bla_{MBL}$-encoding plasmids co-harbored multiple resistance determinants, including serine-β-lactamase for monobactams, 16 S RNA methylases for aminoglycosides, rifampin-modifying enzymes for rifampin, chloramphenicol acetyltransferase for chloramphenicol, esterase for macrolides[2,3,6], thus expanding to multidrug resistance and pandrug resistance.

As a consequence, the treatment of infections by NDM-1 producing clinical strains require the use of last-resort antibiotics e.g., COL[7–9]. COL acts by associating with the anionic lipopolysaccharide (LPS) component of Gram-negative outer membrane, leading to the disruption of the membrane, leakage of intracellular contents and ultimately lytic cell death[10]. Unfortunately, the efficacy of COL has been seriously compromised in the regular treatment of lethal bacterial infections owing to the emergence of MCR-1 enzyme since 2015[11]. Unlike chromosomally mediated polymyxin resistance, which is limited to clonal expansion[12], the plasmid-borne mcr-1 (and its homologues, mcr-2 to −10)[13] is responsible for a transferable mechanism of polymyxin resistance and has been disseminated over 40 countries/regions covering five continents[14]. In addition to conferring resistance to polymyxin antibiotics, mcr-1 could also confer bacterial resistance toward lysozyme[15]. The situation gets exacerbated by the fact that mcr-1 could cotransfer with mcr-3 or mcr-5[16,17] and even coexist with $bla_{MBL}$, including $bla_{NDM-1}$ and its variants $bla_{NDM-5}$ and $bla_{NDM-9}$[18–21]. Moreover, a clinical E. coli isolate has been found to co-carry mcr-1, mcr-3, and $bla_{NDM-5}$ genes[22]. Thus, in clinic context, the co-existence of MBL(s) and MCR(s) in infectious pathogens has raised serious concerns that common infections with these "superbugs" may soon be untreatable, which will severely endanger public health system and leave clinicians with virtually no therapeutic options.

NDM-1, as a typical member of B1 class of MBLs, possesses two Zn(II) ions intercalated by a nucleophile hydroxide in its active site, which is delimited by flexible loop 3 and 10[23]. The two Zn(II) ions have been validated to be crucial for its hydrolytic activity to launch a nucleophilic attack on β-lactam rings in β-lactam antibiotics[24]. Additionally, the presence of Zn(II) ions is beneficial for the accumulation of MBLs in the bacterial periplasm owing to the rapid degradation of these enzymes in their non-metallated forms[25]. MCR-1, a recent member of the phosphoethanolamine (PEA) transferase family, catalyzes the addition of the PEA moiety to the phosphoric group of lipid A moiety on LPS, rendering the bacterial membrane more electropositive and repelling cationic COL[26]. This enzyme contains a Zn(II) cofactor in its active site, coordinating to Glu246, Asp465, His466 and a conserved nucleophilic attack group, phosphorylated Thr285 (TPO285)[27,28]. The Zn(II) ion in the active site of MCR-1 has been validated to be vital for the PEA transfer process; notwithstanding its biological roles in MCR proteins are still controversial[29,30].

Our previous studies showed that an antipeptic ulcer bismuth drug, colloidal bismuth subcitrate, could resensitize MBL-positive bacteria to β-lactam antibiotics through kicking out the Zn(II) cofactors by Bi(III) from the active site of MBLs, thus disrupting their abilities to hydrolyze β-lactam ring in carbapenem[31]. We therefore hypothesize that a metallodrug-antibiotic combination serves as an effective strategy to combat "superbugs" that co-produce MBLs as well as other Zn-dependent resistant determinants, e.g., MCRs. In the present study, we identified an antirheumatic drug, auranofin (AUR, its chemical structure is shown in Supplementary Fig. 1a) as an effective dual inhibitor of MBLs and MCRs. Our results showed that AUR irreversibly inhibit NDM-1 activity and disrupted the function of MCR-1 via displacing Zn(II) ions in the active sites and thus forming Au-NDM-1 or Au-MCR-1. Importantly, AUR synergized with antibiotics on killing a spectrum of bacterial pathogens carrying $bla_{MBL}$ and/or mcr genes and significantly suppressed the resistance development of either MBL or MCR. Notably, AUR potently restored the susceptibility of MCR-1- and NDM-5-co-producing pathogens to COL in murine peritonitis models. This work clearly elucidates the antimicrobial action of a gold(I)-based drug and opens a horizon for the treatment of infections caused by superbugs carrying $bla_{MBL}$ and/or mcr genes.

## Results

**Primary screening identifies auranofin as an antibiotic booster.** A primary screening on a battery of metal compounds was performed to seek for COL and/or carbapenem (exemplified by meropenem, MER) synergism against E. coli CKE, a clinical isolate that co-produced NDM-5 and MCR-1 ($MIC_{COL}$ = 8 μg·mL$^{-1}$, $MIC_{MER}$ = 32 μg·mL$^{-1}$). The growth of E. coli CKE was examined in the presence of selected metal compounds at subinhibitory concentration (50 μg·mL$^{-1}$), and COL (1 μg·mL$^{-1}$) or MER (2 μg·mL$^{-1}$) for 18 h. Our results showed that both cobalt(II) chloride and arsenic(III) trioxide could partially boost efficacies of both MER and COL with ~50% of microbial growth to be inhibited. However, Bi(III) (as bismuth nitrate) could only boost the activity of MER but not COL (Supplementary Fig. 1b, c). To our surprise, both Au(I) (as gold(I) chloride, AuCl) and AUR were observed to markedly boost activities of either MER or COL to the extent that no bacterial growth was observed, suggesting its effectiveness on inhibiting both MBL and MCR-1 activity. In view of well-characterized toxicology and pharmacology of AUR[32], we therefore selected AUR for further exploration of its potential as a dual inhibitor of MBLs and MCRs.

**Auranofin inhibits NDM-1 hydrolytic activity.** Native Zn$_2$-NDM-1 protein was first overexpressed and purified according to previously described method[31] to examine the inhibitory action of AUR towards MBLs. Since AUR is a prodrug and metabolizes to an active species $[Au(PEt_3)]^+$ when transferred across cell membranes[33], we therefore used an AUR analogue, chloro(triethylphosphine)gold(I) $[Au(PEt_3)Cl]$, for all the enzyme-based biophysical studies to mimic its action in cellulo. The hydrolytic activity of NDM-1 against a chromogenic substrate, nitrocefin, was examined in the absence or presence of Au(PEt$_3$)Cl according to a previous report[34]. As shown in Fig. 1a, NDM-1 activity decreased as the Au(PEt$_3$)Cl concentration escalated ($IC_{50}$ = 437.9 ± 29.1 nM), with ~97% activity being inhibited, indicative of excellent inhibitory effect of AUR on NDM-1. From the enzyme kinetics assay, we found that Au(PEt$_3$)Cl leads to an evident decrease in the apparent $V_{max}$ from 19.41 μM·min$^{-1}$ to 3.35 μM·min$^{-1}$ while the $K_m$ value retained almost unchanged at around ~67.3 μM, indicative of a typical noncompetitive inhibition (Fig. 1b). A plot of the inverse of the apparent maximal rates versus the concentration of the inhibitor allowed the inhibition

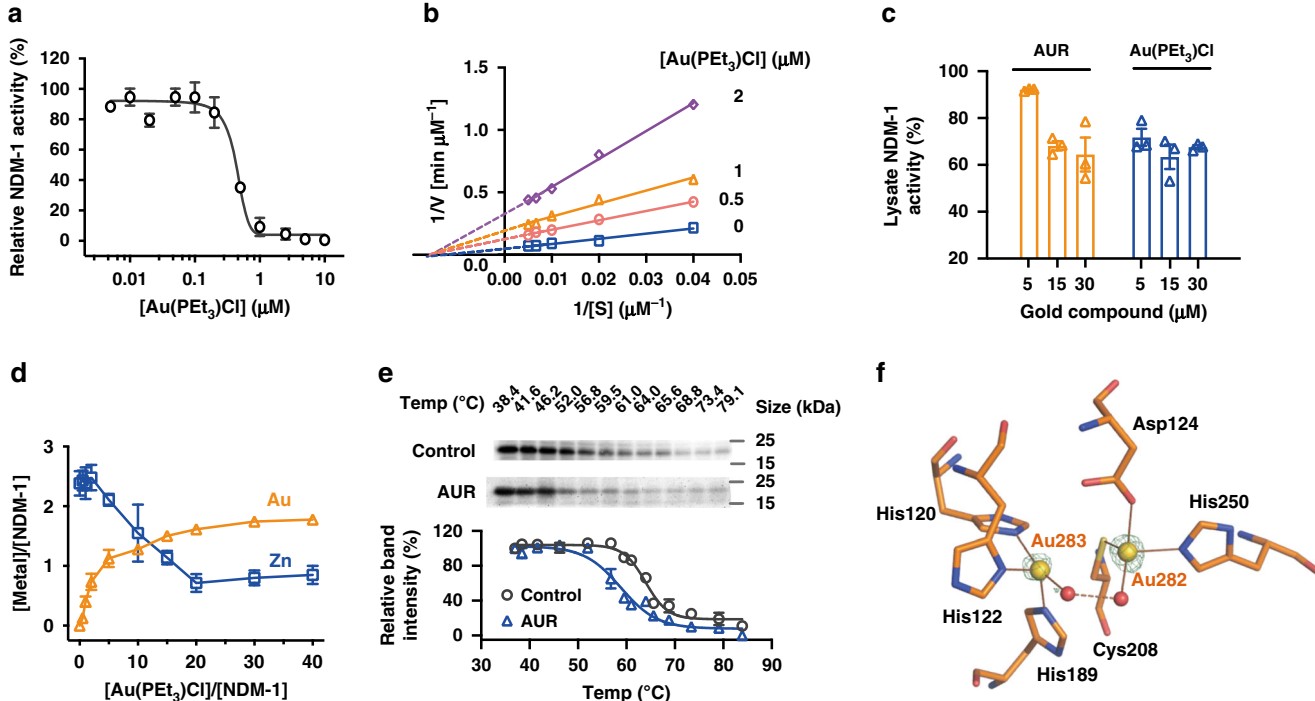

**Fig. 1 Auranofin inhibits MBL activity via the displacement of Zn(II) cofactors. a** Inhibition of NDM-1 activity by Au(PEt$_3$)Cl with IC$_{50}$ of 437.9 ± 29.1 nM. **b** Double reciprocal plot of substrate dependent enzyme kinetics on inhibition of NDM-1 activity by Au(PEt$_3$)Cl. **c** Lysate NDM-1 activity from NDM-Rosetta treated with AUR or Au(PEt$_3$)Cl. **d** The substitution of Zn(II) in Zn$_2$-NDM-1 by Au(PEt$_3$)Cl by equilibrium dialysis. The metal content was determined by ICP-MS. **e** Cellular thermal shift assays showing the binding of Au(I) to NDM-1 in intact NDM-Rosetta cells. NDM-1 melting temperature was shifted from 63.7 to 58.6 °C for control and AUR-treated group, respectively. The images show the representative blottings of three independent experiments. **f** Structure of the active site of Au-NDM-1 (PDB ID: 6LHE) with the anomalous density peak of Au shown as yellow spheres and water molecules as red spheres, and anomalous density peak of Au in green mesh contoured at 7.0σ. **a, c–e** Data are presented as mean values ± SEM, n = 3 biologically independent samples. Source data are provided as a Source Data file.

constant ($K_i$) to be calculated as 389.1 nM for Au(PEt$_3$)Cl. This suggests Au(PEt$_3$)Cl converted NDM-1 into an inactive state rather than competing for the active site with substrates.

It is commonly believed that AUR has a high affinity to thiolate sulfurs (cysteine residues) in proteins to form stable and irreversible adducts[33]. To examine the role of cysteine in NDM-1 inhibition by AUR, we mutated Cys208 to Ala208, i.e., NDM-1-C208A. We found that the hydrolytic activity of the mutant was significantly reduced as demonstrated previously[31,35]. Its activity was inhibited by only ~13.6% upon incubation with Au(PEt$_3$)Cl at 100 μM (Supplementary Fig. 2), indicating that the interaction of AUR with Cys208 is important for its inhibition on NDM-1. Importantly, the lysates extracted from an engineered NDM-1 positive E. coli strain denoted as NDM-Rosetta, that exposed to overnight treatment of AUR (and Au(PEt$_3$)Cl) at escalating concentrations, showed suppressed activity (up to ~30%) towards nitrocefin (Fig. 1c). These data demonstrated that AUR could effectively inhibit NDM-1 activity both in vitro and in cellulo.

**Auranofin displaces Zn(II) cofactors with Au(I) in NDM-1.** To explore the inhibitory role of AUR, we first examined its impact on the Zn(II) cofactors of NDM-1. By using 4-(2-pyridylazo) resorcinol (PAR) assay, we found that c.a. 1.09 molar equivalents of Zn(II) was gradually released from native Zn-NDM-1 upon the addition of Au(PEt$_3$)Cl at 200 μM (Supplementary Fig. 3a, b). In addition, the activity of Zn-NDM-1 was lost in a time-dependent manner when exposed to Au(PEt$_3$)Cl treatment (Supplementary Fig. 3c), indicating that the inhibition of NDM-1 by Au(PEt$_3$)Cl may be attributable to the deprivation of Zn(II). By using

equilibrium dialysis, we showed that the addition of increasing amounts of Au(PEt$_3$)Cl to the native protein resulted in ca. 1.77 ± 0.08 molar equivalents of Au(I) bound to NDM-1, accompanied by ca. 1.53 ± 0.21 molar equivalents of Zn(II) removal (Fig. 1d). The data were fitted by one-site binding Hill plot, which gave rise to the dissociation constant ($K_d$) of 4.03 ± 0.65 μM for Au(PEt$_3$)Cl and the maximal binding capacity ($B_{max}$) of 1.95 ± 0.10, indicating that two Au(I) ions bound per monomer of NDM-1. Significantly, the inhibition of NDM-1 activity by Au(PEt$_3$)Cl could not be reversed as the supplementation of 30 molar equivalents of Zn(II) led to less than 20% activity being restored for Au-NDM-1, owing to the limited ability of excess Zn(II) to replace Au(I) (Supplementary Fig. 4).

We next examined the binding of AUR to NDM-1 by matrix assisted laser desorption ionization time of flight mass spectrometry (MALDI-TOF MS) (Supplementary Fig. 5). A peak observed at m/z of 25925.4 could be assigned to NDM-1 monomer. After 18-h co-incubation with AUR at molar ratio of 1:10, new peaks at m/z of 26124.3, 26240.7, 26323.1, and 26442.9 appeared and could be assigned to [NDM-1 + Au], [NDM-1 + Au(PEt$_3$)], [NDM-1 + 2Au)], and [NDM-1 + Au + Au(PEt$_3$)], respectively. These results suggested that [Au(PEt$_3$)]$^+$ binds to NDM-1 via exchange triethlphosphine with the enzyme. The cellular engagement of NDM-1 by AUR was examined by cellular thermal shift assay (CETSA)[36]. The overnight exposure to AUR (12 μg·mL$^{-1}$) treatment decreased the cellular thermal stability of NDM-1 by $\Delta T_m$ = 5.2 °C in NDM-Rosetta, indicative of the binding of AUR to NDM-1 in intact cells. (Fig. 1e).

We further explored the binding mode at atomic level by X-ray crystallography. Au-bound NDM-1 was first obtained by

incubating apo-NDM-1 with 10 molar equivalents of AUR overnight at 25 °C and then co-crystallized using a typical sitting-drop vapor diffusion method. The crystal structure of Au-NDM-1 was determined at 1.20 Å resolution (PDB ID: 6LHE) and the binding of Au(I) to the protein was confirmed by X-ray excitation spectrum, which solely showed the excitation peak for gold (Au-L3) at the radiation energy around 9.8 keV, while no peak for zinc around 8.6 keV was observed (Supplementary Fig. 6a). Super-imposition of all its Cα atoms with Zn-NDM-1 (PDB ID: 5ZGE) showed no significant overall conformational changes (RMSD = 0.211 Å) (Supplementary Fig. 6b, c). As shown in Fig. 1f, two Au ions, *viz*, Au$^{282}$ and Au$^{283}$ were clearly observed in the active site, tetrahedrally coordinated to Cys208, His250, Asp124, a water molecule (w$^{291}$), and His122, His120, His189, and a water molecule (w$^{410}$), respectively. The distance between two Au ions (~3.8 Å) is much shorter than that between two Zn ions (~4.6 Å) in the intact NDM-1 crystal[23]. An additional Au ion (Au$^{281}$) resided in the interface of two protein monomers coordinating to Asp223, Glu152, a water molecule, and a Glu227 from an adjacent NDM-1 molecule in a distorted tetrahedral geometry (Supplementary Fig. 6d, e). The bond lengths of Au(I) with the side-chains of those amino acid residues were overall in the range of 1.9–2.5 Å and generally comparable to those for Zn(II) (Supplementary Tables 1, 2). The occupancies of the three Au ions were *ca*. 0.5 with the Au$^{283}$ in the Zn1 site showing slightly lower occupancy (0.45). The active-site pocket of NDM-1 in the crystal structure was widely open and two Au ions resided in the bottom of the cavity, apparently being assessable to AUR. Overall, the data agreed well with our biophysical characterization that the Zn-dependent hydrolysis function of NDM-1 was inhibited by AUR through displacement of the Zn(II) in the active.

**Auranofin resensitizes MBL-positive Enterobacteriaceae to carbapenem**. We then examined whether AUR could enhance the antimicrobial activity of MER against an NDM-1 positive *E. coli* isolate, NDM-HK (MIC$_{MER}$ = 16 μg·mL$^{-1}$)[37]. The MIC value of MER was remarkably reduced by 64 folds in the presence of 16 μg·mL$^{-1}$AUR, for which the fractional inhibitory concentration index (FICI) was estimated to be 0.156, i.e., indicative of synergy (FICI ≤ 0.5) (Fig. 2a). Importantly, the combination only led to additive killing of MBL negative strain with FICI of 0.625 (0.5 < FICI < 1 indicates addition) during the same assay operations (Fig. 2b). The AUR-MER combination exhibited potent bacter-icidal effect as revealed by time-kill assay. Early-log phase culture of NDM-HK (10$^6$ CFU·mL$^{-1}$) was subjected to the treatments of vehicle, MER, AUR or their combination; and the bacterial growth was measured at different intervals by agar plating. As expected, no evident CFU reduction of NDM-HK was observed among untreated, MER, and AUR group by 24 h. In contrast, the bacterial loads in the combination group plummeted by more than five orders than any single component and their outgrowth were prevented throughout 24-h exposure (Fig. 2c). Furthermore, the antimicrobial potency of AUR-MER combination could be profiled against a panel of Enterobacteriaceae that produced either B1 MBL (VIM-2, IMP-4) or B2 MBL (CphA) (Fig. 2d), with FIC index ranged from 0.133 to 0.375 (Supplementary Table 3). This confirms that AUR serves as a broad-spectrum inhibitor of MBLs. Resistance levels to MER were greatly elevated upon exposure to MER at subinhibitory concentrations, as its MIC value increased by 8-folds over a period of 16 serial passages (Fig. 2e). Importantly, the resistance level was slightly increased by 2-folds upon the combined use of AUR and MER. In addition, in the presence of AUR (30 μg·mL$^{-1}$), the mutation prevention index (MPI) of MER significantly decreased from 32 to 1 (Fig. 2f). Our combined data clearly demonstrate that AUR restores the

carbapenem activity against MBL-positive bacteria and restricts the enrichment of mutant subpopulation in them.

**Auranofin disrupts action of MCR-1 catalysis**. We next inves-tigated the feasibility of AUR as a resistance breaker of MCR-1, which is devoid of cysteine residues in its active site. MCR-1 is organized into two domains including an N-terminal inner membrane-bound domain and a soluble, periplasmic domain equipped with a Zn-dependent catalytic core and two (putative) substrate-binding pockets[38]. This enzyme could bind PEA and lipid A in respective pockets and launch the Zn-dependent transfer of PEA to lipid A with the assistance of its transmem-brane domain[38]. We asked whether AUR could functionally disrupt MCR-1 in vitro. To address this, the (semi-)activity of MCR-1 was measured by a thin layer chromatography (TLC)-based method using a fluorescently labeled natural mimetic substrate, nitrobenzodiazole-labeled glycerol-3-phosphoethanolamine (NBD-glycerol-3-PEA)[39]. First, we pur-ified the full-length membrane protein Zn-MCR-1 with a gradient concentration of the detergent n-dodecyl-β-D-maltoside (DDM) to preserve its catalytic activity. The protein was incubated with or without 10 molar equivalents of Au(I) compounds overnight and subsequently mixed with NBD-glycerol-3-PEA, followed by TLC detection. Zn-MCR-1 could cleave PEA group from NBD-glycerol-3-PEA, and the loss of PEA group led to the faster migration of NBD-glycerol as revealed in Fig. 3a. In contrast, no observable migration relative to either the substrate only (as a control), AuCl or Au(PEt$_3$)Cl treatment group was found (Fig. 3a), indicative of the inhibition of cleavage activity of MCR-1 by Au(I).

Given that membrane potential serves as a good indicator for the proper function of MCR-1[40], we further explored if AUR could disrupt the action of MCR-1 in intact cells. The membrane potentials of MCR-1 positive or negative *Shigella flexneri* (*S. flexneri*) was determined by using diethyloxacarbocyanine [DiOC$_2$(3)], a green fluorescent dye that formed red fluorescent aggregated with an increase in membrane potential, upon the treatment of AUR, Au(PEt$_3$)Cl, and carbonyl cyanide 3-chlorophenylhydrazone (CCCP, as a positive control), respec-tively. As shown in Fig. 3b, green/red fluorescent ratios decreased from 6.14 to 4.08, indicative of the significant reduction of the membrane negative charge in MCR-1 positive *S. flexneri* in comparison to the MCR-1 negative strain. Importantly, negligible reduction in negative charge was observed for MCR-1-positive *S. flexneri* that were treated either by AUR (3 μM), Au(PEt$_3$)Cl (3 μM), or CCCP (5 μM) (Fig. 3b). This suggests that AUR effectively prevents the MCR-1-induced loss of negative charges in cellulo.

**Auranofin inhibits MCR-1 activity through displacement of Zn (II) cofactor**. To investigate the inhibitory effect of AUR, we first overexpressed and purified the soluble, periplasmic domain of truncated MCR-1 (residues 201-541, denoted as MCR-1-S), which was determined to bind three equivalents of Zn(II) ions by ICP-MS. We showed by PAR assay that titration of Au(PEt$_3$)Cl (10 molar equivalents) to the Zn-MCR-1-S led to *ca*. 1.98 molar equivalents of Zn(II) to be released from the protein rapidly within 30 min (Supplementary Fig. 7). By equilibrium dialysis, we found unexpectedly that Au(I) ions bind intact MCR-1-S (Zn-MCR-1-S) with the binding affinities ($K_d$) of 3.36 ± 0.38 μM for Au(PEt$_3$)Cl and binding stoichiometry ($B_{max}$) of 3.00 ± 0.60. Zn (II) content in Zn-MCR-1-S was decreased gradually and *ca*. 2.21 molar equivalents of Zn(II) was removed ultimately during the exposure to Au(PEt$_3$)Cl (Fig. 3c). Notably, the supplementation of up to 5 molar equivalents of Zn(II) could not displace Au(I)

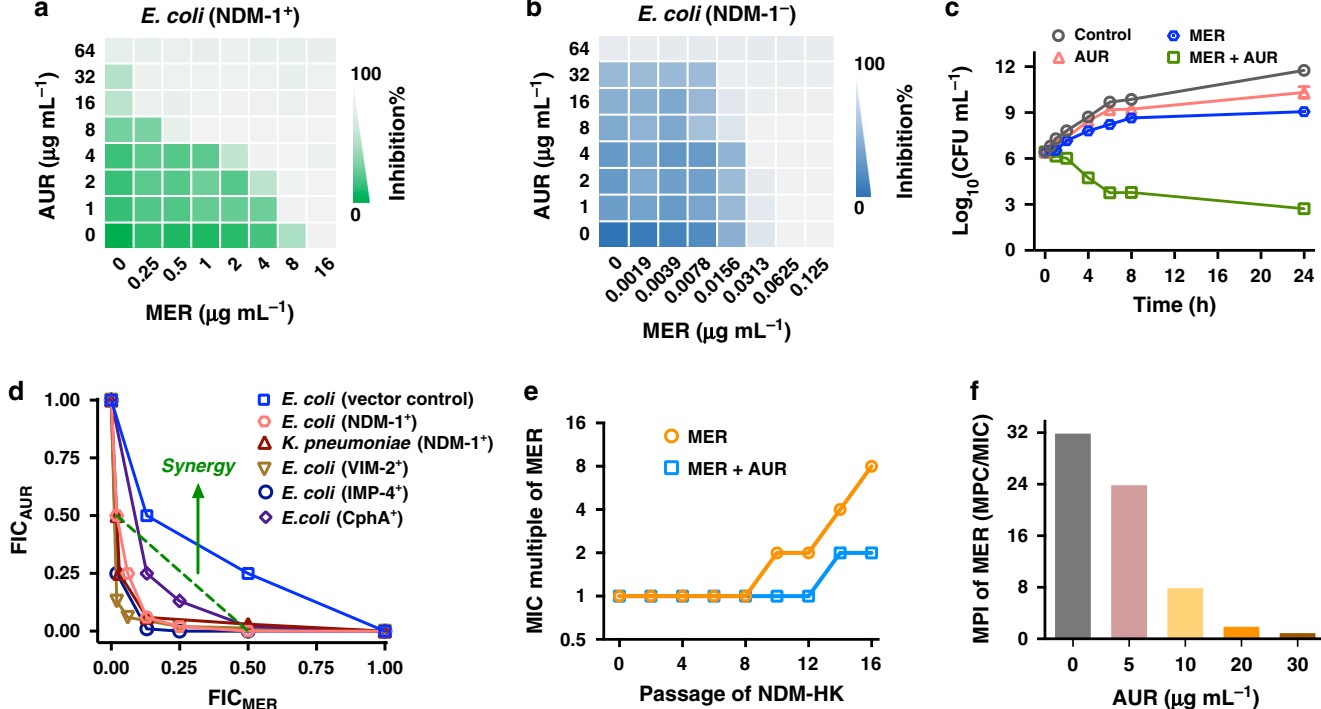

**Fig. 2 Auranofin restores the susceptibility of MBL-positive bacteria to carbapenem. a, b** Representative heat plots of microdilution checkerboard assays for the combination of MER and AUR against (**a**) NDM-1-positive *E. coli* and (**b**) NDM-1-negative *E. coli*. **c** Time-kill curves for MER or AUR monotherapy or their combination therapy against NDM-HK during 24-h incubation. The concentrations of MER and AUR are used at 16 μg·mL$^{-1}$ and 30 μg·mL$^{-1}$, respectively. Data are presented as mean values ± SEM, $n = 3$ biologically independent samples. **d** Isobolograms of the combination of MER and AUR against different MBL-positive bacterial strains. The green dash line indicates ideal isobole, where drugs act additively and independently. Data points below this line indicate synergism. **e** Resistance acquisition curves during serial passage with the subinhibitory concentration of MER or the combination of MER and AUR against NDM-HK. MIC test was performed every two passages. **f** Bar chart showing MPI indices of MER in the presence of increasing concentration of AUR against NDM-HK. **a, b** Data in **a** and **b** represent the mean OD$_{600}$ of two biological replicates. Source data are provided as a Source Data file.

from Au-MCR-1-S (Fig. 3d). These data suggest that AUR disrupts the function of MCR-1 via displacement of Zn(II) by Au(I). We next investigated cellular engagement of MCR-1-S by AUR using CETSA, the thermal stability of MCR-1-S was decreased by *ca*. 3.6 °C ($\Delta T_m$) upon treatment of AUR (15 μg·mL$^{-1}$) in MCR-1-S-producing *E. coli* (denoted as MCR-1-S-BL21) (Fig. 3e), confirming the binding of AUR to MCR-1 in cellulo.

To further unveil the detailed inhibition mode, we first prepared a Zn-bound crystal of MCR-1-S (PDB ID: 6LI4) and then transformed to Au-bound MCR-1-S crystal (PDB ID: 6LI6) by a soaking method[31]. The crystal structure of Au-MCR-1-S was resolved at 1.68 Å resolution and the binding of Au(I) to the MCR-1-S was also confirmed by X-ray excitation spectrum (Supplementary Fig. 8). Superimposition of this crystal structure with the Zn-bound or apo-bound MCR-1-S (PDB ID: 6LI5) overall C$_a$ atoms showed negligible overall conformational change with RMSD values of 0.476 and 0.573 Å, respectively (Supplementary Fig. 9). From the density map, three Au ions were clearly visible in one chain in an asymmetric unit, and a single Au ion (Au$^{542}$ with occupancy of 0.5) was found in the Zn-dependent catalytic core, coordinating with four residues, i.e., Glu246, Asp465, His466, and TPO285 to form a distorted tetrahedral geometry (Fig. 3f). The bond lengths of Au(I) with the side-chains of amino acid residues are overall slightly longer than those for Zn(II) (Supplementary Tables 4, 5), probably owing to the larger ionic radius of Au(I) (1.37 Å) than Zn(II) (0.74 Å). The second Au ion (Au$^{543}$) linearly coordinated to Nε2-His252 and PEt$_3$ group (N–Au–P angle 171.5°) (Supplementary Fig. 10). The third Au ion (Au$^{544}$) coordinated to Nε2-His424 and a water molecule

in a quasi-linear geometry (N–Au–O angle of 159.0°) (Supplementary Fig. 10). The Au ion (Au$^{543}$) was only observed in one chain in the asymmetric unit with occupancy of ~0.3, but not in the other one (Supplementary Fig. 10a), inferring the less accessibility of Au(PEt$_3$)Cl to this site. The crystallographic analysis demonstrates that AUR binds to MCR-1 and displaces Zn(II) in the active site through Au(I).

**Auranofin resensitizes MCR-positive Enterobacteriace to colistin.** By using checkerboard microdilution assay, we observed a synergistic pattern that AUR (5 μg·mL$^{-1}$) resensitized MCR-1-J53 to COL by 64 folds, with its MIC dropped from 8 μg·mL$^{-1}$ to 0.125 μg·mL$^{-1}$ (Fig. 4a). This combination showed typical synergism with FIC index of 0.125 for MCR-1-J53 but not for MCR-1-negative strain (FIC = 0.531, Fig. 4b). Time-kill assay revealed that AUR (6 μg·mL$^{-1}$) and COL (2 μg·mL$^{-1}$) combinedly presented a potent bactericidal activity as reflected by over 10$^7$ decreases in viable MCR-1-J53 compared with the control or any single components after 24-h exposure (Fig. 4c). Furthermore, the two drugs synergized to kill a spectrum of MCR-positive bacterial pathogens with the MIC of COL above susceptible breakpoint (2 μg·mL$^{-1}$) and FIC index ranging from 0.125–0.281 (Fig. 4d and Supplementary Table 6). Under identical conditions, AUR (0.625 μg·mL$^{-1}$) significantly decreased the MIC of COL against an *E. coli* strain produced different MCR variants including 5 MCR-1 variants and 6 MCR homologs, by 8–16-folds (Fig. 4e), implying the broad-spectrum antimicrobial potency of this combination against MCR(s)-positive bacteria.

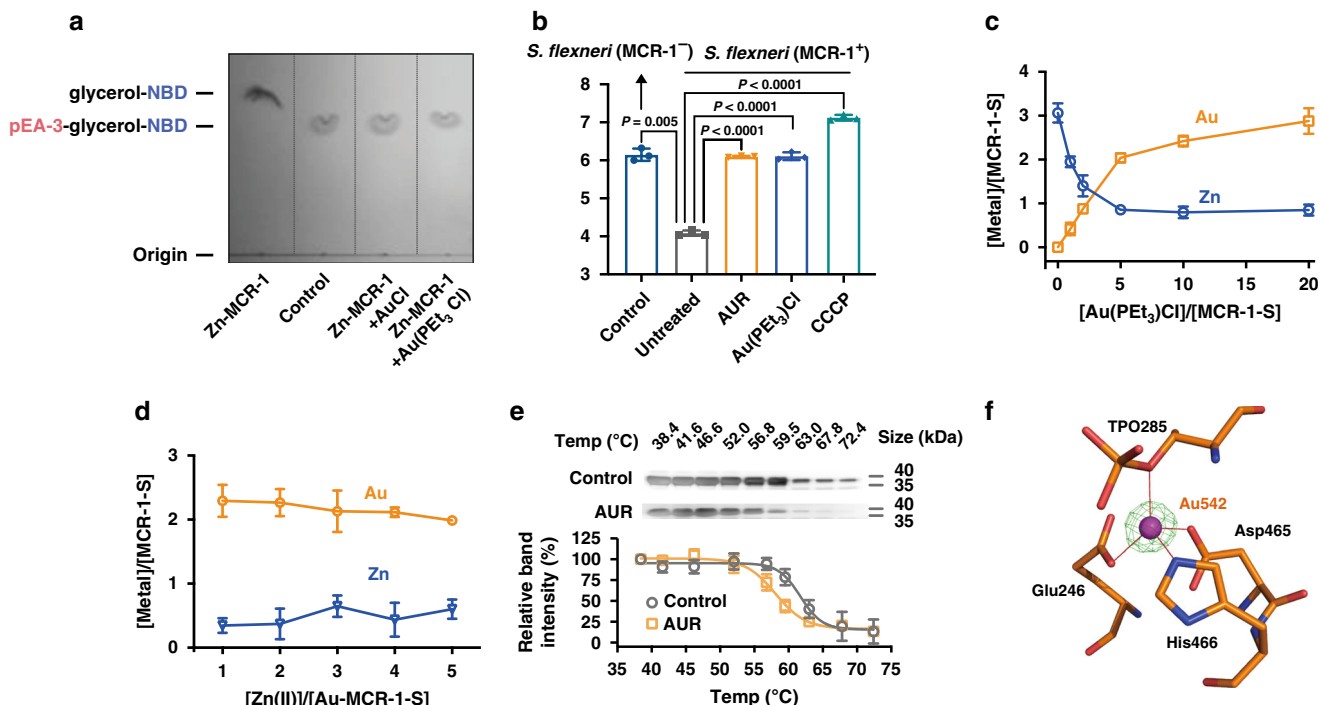

**Fig. 3 Auranofin disrupts the Zn(II)-dependent function of MCR-1. a** Inhibition of MCR-1 cleavage activity on NBD-glycerol-3-PEA by AuCl and Au(PEt₃)Cl. A representative image of TCL plate is shown here. **b** Membrane potential changes upon AUR treatment in MCR-1 positive and negative *S. flexneri* as determined by the ratios of green to red fluorescent signals. **c** Substitution of Zn(II) in Zn₃-MCR-1-S by Au(PEt₃)Cl over equilibrium dialysis. The metal content was determined by ICP-MS. **d** The metal contents upon supplementation of various amounts of Zn(II) into Au-MCR-1-S over equilibrium dialysis. The metal content was determined by ICP-MS. **e** Cellular thermal shift assays showing the binding of Au(I) to catalytic domain of MCR-1 in intact MCR-1-S-BL21 cells. MCR-1-S melting temperature was shifted from 61.7 to 58.1 °C for control and AUR-treated group, respectively. The images show the representative blottings of three independent experiments. **f** Structure of the active site of Au-MCR-1-S (PDB ID: 6LI6) with the anomalous density peak of Au ion shown as a purple sphere and anomalous density peak of Au in magenta mesh contoured at 5σ. **b**–**e** Data are presented as mean values ± SEM, *n* = 3 biologically independent samples. **b** *P* values were determined by an unpaired two-tailed student *t*-test with Welch's correction. Source data are provided as a Source Data file.

The in vitro serial passage was also used to examine whether AUR could suppress the resistance development in MCR-1 positive bacteria. We showed that the MIC values of COL increased by 100 folds after a period of 16 serial passages when used alone (Fig. 4f). In contrast, the MIC values of COL was increased only by 4 folds at the end of experiment upon the combined use of AUR. Similar to the result of MPC assay for NDM-1, AUR decreased the mutation prevention indices (MPI) of COL by 32 folds, i.e., from 16 to 0.5 in a dose-dependent manner (Fig. 4g), suggesting that AUR serves to diminish the occurrence of higher level of resistance in Enterobacteriaceae.

**Auranofin restores in vivo efficacy of colistin.** To further evaluate the potential utility of combination regimens, we assessed AUR in combination with COL in vivo using two murine models of MCR-1- and -NDM-5-positive *E. coli* and MCR-1-positive *K. pneumoniae* infections. AUR was dosed at subinhibitory concentrations, which exhibited synergistic potentiation based on in vitro assessing (Figs. 2, 4, Supplementary Tables 3, 6), and were below the toxicity dosage according to a pilot study. COL was administered at approximately human equivalent dose (daily dosage: 2.5—5.0 mg·kg⁻¹ [41]). Balb/c mice were first systemically infected intraperitoneally (*i.p.*) with a sublethal dose of *K. pneumoniae* (MCR-1⁺) (~2 × 10⁶ CFU per mouse), and then administrated with single dose of vehicle, AUR (0.5 mg·kg⁻¹), COL (2 mg·kg⁻¹) or their combination 1-h post-infection, respectively. The bacterial loads in the liver and spleen were barely affected by

AUR alone and the isolate showed certain extent of resistance to COL monotherapy after 48 h. In contrast, the bacterial loads of COL-AUR combination group plummeted by over 10 folds in the spleen and liver than that of COL alone (Fig. 5a, b). In a separated model, groups of Balb/c mice were *i.p.* injected with lethal dose of *E. coli* CKE, and received single dose of AUR, COL or their combination 0.5-h post-infection, respectively. We found that both vehicle control and AUR alone led to the death of all mice in respective group within 3 days, and COL monotherapy failed to rescue four out of six infected mice till the endpoint of experimental period. Remarkably, all mice were survived upon the combination therapy at 5 days following infection (Fig. 5c), highlighting the in vivo effectiveness of AUR-COL combination against MCR-1- and -NDM-5-positive bacterial infections.

## Discussion

The dissemination of MBL-producing Enterobacteriaceae, as well as the acquisition of polymyxin resistance gene *mcr-1* in those bacteria, may lead to the emergence of untreatable bacterial infections, thus posting significant threat to healthcare systems globally. With little remission from the therapeutic reliance on the current pipeline of β-lactam antibiotics and COL, the combination therapy consisting of an antibiotic resistance and an antibiotic breaker offers promising options to narrow the gap between pan-resistant bacteria and the development of new antibiotics. In this connection, recent studies have reported a number of mechanism-based or structure-based inhibitors of NDM-1 and other MBLs, primarily including covalent inhibitor

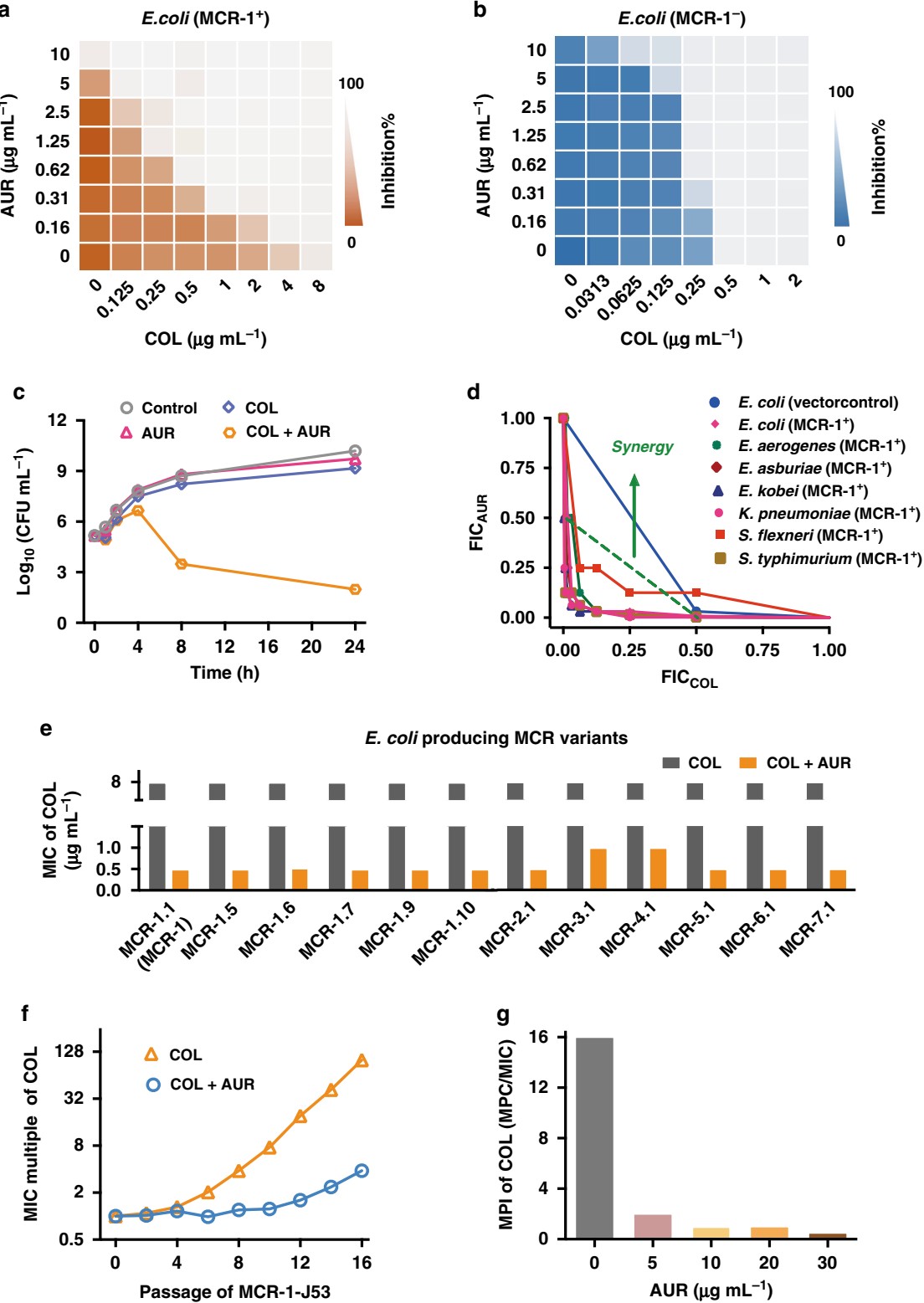

(β-lactam and non–β-lactam inhibitor), and non-covalent inhibitors (e.g., zinc chelate and coordinating agent)[42]. In parallel, other strategies have also been developed to restore COL sensitivity to MCR-1-postivie bacteria, by combined use with other antimicrobial agents (e.g., clarithromycin[43]), or substrate mimickers (e.g., ethanolamine[44], D-glucose[44]). However, owing to the vast differences in the active sites and modes of action between MBLs and MCRs, no inhibitors are available clinically to

inhibit both enzymes simultaneously. Here, we propose AUR as a promising dual inhibitor of MBL and MCR enzymes. AUR serves as an antibiotic adjuvant to resensitize MBL- and/or MCR-positive bacteria to carbapenem and COL, and slows down the development of higher level of resistance. Importantly, the combination of AUR and COL showed therapeutic potency for the treatment of infections caused by carbapenem-and poly-myxin-resistant bacteria at subclinical doses in vivo.

**Fig. 4 Auranofin synergizes with colistin to kill MCR-positive bacterial strains. a, b** Representative heat plots of microdilution checkerboard assay for the combination of COL and AUR against **a** MCR-1-positive *E. coli* and **b** MCR-1-negative *E. coli*. **c** Time-kill curves for COL and AUR monotherapy or combination therapy against MCR-1-J53 during 24-hr incubation. The concentrations of COL and AUR are 2 µg·mL⁻¹ and 6 µg·mL⁻¹, respectively. Data are presented as mean values ± SEM, $n = 3$ biologically independent samples. **d** Isobolograms of the combination of COL and AUR against different MCR-1-positive bacterial strains. The green dash line indicates ideal isobole, where drugs act additively and independently. Data points below this line indicate synergism. **e** Bar charts showing the reduction of COL MIC for *E. coli* J53 that produced MCR-1 variants and MCR homologs in the combined use with AUR at fixed concentration of 0.625 µg·mL⁻¹. **f** Resistance acquisition curves during serial passage with the subinhibitory concentration of COL or the combination of COL and AUR against MCR-1-J53. MIC test was performed every two passages. **g** Bar charts showing MPI indices of MER in the presence of increasing concentration of AUR against MCR-1-J53. **a, b** Data in **a** and **b** represent the mean OD₆₀₀ of two biological replicates. Source data are provided as a Source Data file.

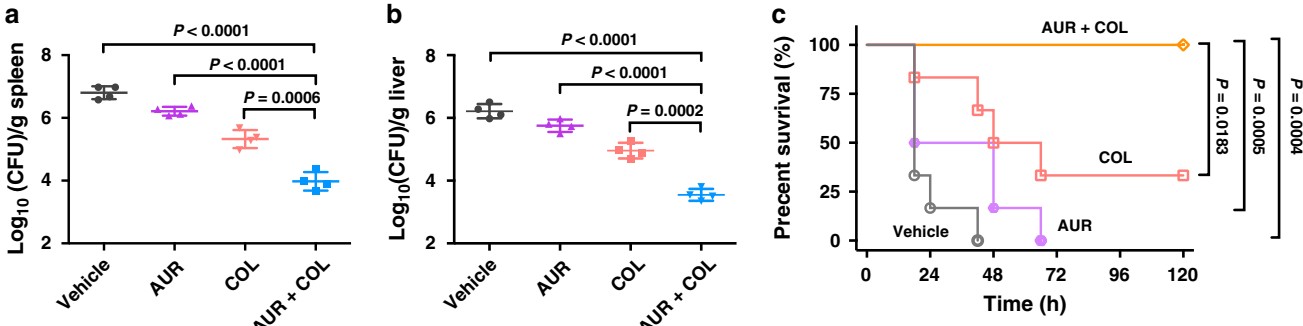

**Fig. 5 The combination of AUR and COL shows potency in vivo. a, b** Balb/c mice were given a sublethal dose of *K. pneumoniae* 9607 (MCR-1⁺) and received single dose of *i.p.* administration of vehicle, AUR, COL or their combination ($n = 4$ per group). Bacterial loads in the spleen (**a**) and liver (**b**) are shown. **c** Survival curves showing combination efficacies in the peritonitis infection model. Balb/c mice were infected by a lethal dose of *E. coli* CKE (MCR-1⁺, NDM-5⁺) and received single dose of *i.p.* administration of vehicle, AUR, COL or their combination ($n = 6$ per group). **a, b** Error bars represent mean ± SEM for biological replicates. **a, b** $P$ values were determined by an unpaired two-tailed student *t*-test with Welch's correction. For **a** $P < 0.0001$, significance difference between AUR + COL and Vehicle or AUR group, $P = 0.0006$, significance difference between AUR + COL and COL. For **b**, **a** $P < 0.0001$, significance difference between AUR + COL and Vehicle or AUR group, ***$P = 0.0002$, significance difference between AUR + COL and COL. **c** $P$ values were determined by two-tailed Log-rank (Mantel–Cox) test. $P = 0.0004$, significance difference between AUR + COL and vehicle group, $P = 0.0005$, significance difference between AUR + COL and AUR group, $P = 0.0183$, significance difference between AUR + COL and COL group. Source data are provided as a Source Data file.

Although the exact molecular mode of action of AUR has not been fully understood, it is believed to be associated with its strong thiophilic nature. In general, AUR metabolizes into a [Au (PEt₃)]⁺ species via an exchange of its tetraacetylated thioglucose ligand with cellular or blood thiols, and then targets cysteine residues (and/or histidine residues), or redox-active selenocysteine through thiolate exchange[45]. In spite of occupying similar positions with Zn(II) in both enzymes, Au(I) normally prefers linear coordination (two-coordination) with four-coordination being less frequent and hardly has five-coordination. Ligand exchange in Au(I) complex is much slower than that of Zn(II), particularly for tetrahedral Au(I) complexes[46]. Moreover, the differences and affinities towards water between Zn(II) and Au(I) may largely alter the activated energy of transition state and thus makes the catalytic process no longer work under the same condition, leading to inhibition of the enzymes upon Zn(II) displacement by Au(I).

Our combined data have demonstrated that AUR could break the resistance mediated by B1 and B2 class MBLs via targeting the cysteine residue in the active site of those enzymes. The emanation of [Au(PEt₃)]⁺ from AUR and its binding to NDM-1 could be well characterized by the MALDI-TOF MS data (Supplementary Fig. 4). Our crystallographic analysis confirmed the binding of Au(I) species to the cysteine residue at the active site of NDM-1 (Fig. 1f). Unlike bismuth[31], each AUR displaces one Zn (II) and tetrahedrally coordinated to respective three residues in either Zn1 or Zn2 site of NDM-1. The loss of inhibitory activity of AUR for C208A NDM-1 mutant confirmed the importance of cysteine during the engagement of AUR to NDM-1

(Supplementary Fig. 3b). This feature suggests that AUR could be used as a broad-spectrum inhibitor of B1 and B2 class of MBL to efficiently resensitize carbapenem-resistant bacteria to conventional antibiotics, as revealed by our susceptibility testing data (Fig. 2 and Supplementary Table 3).

Previous studies have shown that all the members of the MCR family proteins possess an identical secondary structure, with a high degree of conservation for the amino acid residues in the active site and PEA-interacting cavities. In particular, residues Glu246, Thr285, His390, Asp465, and His466 in the zinc-binding core and residues Asn108, Thr112, Glu116, Ser330, Lys333, His395, and His478 in the PEA binding cavity in MCR-1 are also located in similar positions in other MCR members[47]. Although there is no cysteine residues in the active site of MCR enzyme, to our surprise, unlike Bi(III) drugs, AUR also targets MCR-1 protein at its zinc-catalytic core through the side-chains (εN) of histidine residues, suggesting that the "softness" of histidine (e.g., εN) is tunable by a protein. Indeed, the unique chemistry of metal ions has recently aroused great interests in using metalloagents for combating antimicrobial resistance[48]. We show that the Zn (II) ions in MCR-1 could be irreversibly kicked out by AUR (Fig. 3c, d) and form an Au-bound MCR-1 with Au(I) coordinating to Glu246, Asp465, His466, and TPO285 (Fig. 3f). As a consequence, AUR functionally disrupts MCR-1 in its cleavage action of PEA from the substrate, resulting in prevention of MCR-1 from perturbing the negative charges on bacterial membrane (Fig. 3b, c). Importantly, AUR was found to restore the susceptibility of bacteria carrying *mcr-1* gene or its variants/ homologs to COL at concentrations below clinical breakpoint

(resistant breakpoint > 2 µg·mL$^{-1}$ for Enterobacteriaceae), indicative of its broad-spectrum inhibitory activity against MCR family proteins (Fig. 4a–e). We hypothesize that, upon the inhibition of MCR-1 by AUR on cell membrane, COL may act as a membrane-disrupting agent[49] to in turn facilitate the uptake of hydrophobic AUR in Gram-negative bacteria. Given that the MCR members are functional, unified, and equivalent, these findings would constitute a structural and mechanistic paradigm for further development of MCR-1 inhibitors to circumvent COL resistance.

Despite the rapid evolution of either NDM-1 or MCR-1 into their variants with higher resistant level[50,51], we demonstrate that AUR could suppress the resistance development (Figs. 2e and 4f). The inability to generate a resistant mutant is likely due to the fact that AUR has multiple targets or possesses a nonspecific mode of action in E. coli[52]. As shown in Figs. 2f and 4g, in the presence of AUR (30 µg·mL$^{-1}$), the mutation prevention indices (MPI) of MER and COL were decreased by 32 folds, suggesting that the combined use of AUR restricted the enrichment of mutant subpopulation. The area under the curve (AUC$_{24}$)/MPC and peak plasma concentration (C$_{max}$)/MPC values were considered to be two more accurate predictive indices of activity against resistant mutant than AUC$_{24}$/MIC and C$_{max}$/MIC, with optimal values of >25 and >2.2, respectively. Based on our data, AUC$_{24}$/MPC and C$_{max}$/MPC could be preliminarily estimated to be 29.2 and 2.9 for MER (MERREM® IV, meropenem for injection, for intravenous use, FDA, 1996) and 15 and 0.63 for COL (calculated with steady-state COL AUC$_{ss,0-24}$ of 60 µg·h·mL$^{-1}$ and C$_{ss, average}$ of 2.5 µg·mL$^{-1}$)[53], respectively, demonstrating that the combined use of AUR may serve to lower the emergence of higher-level resistant mutants in clinic.

Notably, the therapeutic potential has been demonstrated by our in vivo data that AUR-COL combination is highly effective in eradicating MCR-1-positive bacteria in peritonitis infection model. Co-administration with AUR (0.25-0.5 mg·kg$^{-1}$) restores the in vivo efficacy of COL, with over 10-fold reduction in K. pneumoniae loads in mouse liver and spleen (Fig. 5a, b), as well as completely prevents the death of the group of mice infected by MCR-1- and NDM-5-positive E. coli (Fig. 5c). AUR was approved by the FDA as an oral anti-inflammatory aid in the treatment of rheumatoid arthritis[32] with well documented with no carcinogenicity, no side-effects, or no other long-term safety concerns[32], and is now undergoing a phase II clinical trial for the treatment of amebiasis or giardiasis[54]. At the currently FDA-approved long-term dose (6 mg per day), the steady-state blood Au concentration would be achieved at 3.5 µM in a 12-week treatment course[52]. Although the action on other Zn enzymes and potential toxicity of auranofin and Au(I)-based compounds are still need further investigation, considering its well-recorded safety in human, AUR as a dual inhibitor of MBLs and MCRs would largely broaden the therapeutic options in treating the infections caused by MCR-1 positive CRE.

Collectively, we have demonstrated that AUR acts synergistically with β-lactam antibiotics and COL on killing a broad spectrum of carbapenem- and/or COL-resistant strains, and slows down the development of β-lactam and COL resistance. Importantly, the in vitro antimicrobial potency of AUR and antibiotics could be well-translated in vivo. Our work clearly elucidates the antimicrobial action of the gold(I)-based drug and opens a horizon for the treatment of infections caused by superbugs carrying bla$_{MBL}$ and/or mcr genes.

## Methods

**Chemicals and bacterial strains**. Meropenem was purchased from TCI Chemicals (Shanghai, China). Kanamycin sulfate and Luria–Bertani (LB) broth powder were purchased from Affymetrix. Auranofin (AUR) was purchased from MedChemExpress (MCE, USA). The fluorescent substrate NBD-glycerol-3-PEA was purchased from Avanti Lipids, USA. TLC plate (Silica gel 60 F$_{254}$, Aluminium sheets, 20 × 20 cm) was purchased from Merck, Germany. All the other chemicals were purchased from Sigma-Aldrich unless otherwise stated. All the sources of bacteria employed for cell-based and animal studies are listed in Supplementary Table 7. All the specific primers for different genes listed in Supplementary Table 8 were synthesized by BGI Gene Company (Shenzhen, China).

**Initial screening of antimicrobial activity on antibiotics and different metals**. Metal salts used for the screening involved cobalt chloride (Co(II)), sodium antimonide (Sb(III)), nickel chloride (Ni(II)), arsenic trioxide (As(III)), copper sulfate (Cu(II)), gold chloride (Au(I)), bismuth nitrate (Bi(III)). Briefly, overnight culture of E. coli CKE was performed 1:1000 dilution and regrew to mid-log phase, then diluted to OD$_{600}$ 0.05–0.1 in each well of 96-well plate. The bacterial suspension was exposed to the treatment of either COL (1 µg·mL$^{-1}$) or MER (2 µg·mL$^{-1}$) in the absence or presence of different metal ions at a fixed concentration (50 µg·mL$^{-1}$). The bacterial growth inhibition was examined by monitoring OD$_{600}$ for 18 h. Wells with no antibiotics or metal compounds served as growth controls and wells with LB medium served as background controls.

**Construction of plasmids**. Plasmids with full-length mcr-1 gene and truncated mcr-1 gene encoding soluble MCR-1 fragment (200-541aa, denoted as MCR-1-S) were from our own collection. The original genes were subcloned into pCR-XL-TOPO and pET-15b vectors via the SacI/BamHI and NdeI/BamHI restriction sites, respectively. Other mcr genes were synthesized by BGI (China) and subcloned into pET-28a via NcoI/XhoI restriction sites, except for mcr-6.1 via NcoI/BamHI site. All the modified plasmids were constructed by using these plasmids as PCR template. PCR was performed using KOD Hot Start DNA polymerase (Novagen) based on the reaction conditions described in the protocol by the manufacturers. Digested fragments with full-length or truncated mcr-1 were inserted into mutated pET-15b via the NdeI/BamHI restriction site for protein overexpression and thermal stability analysis. All the restriction enzymes and T4 ligase were purchased from New England Biolabs (UK) Ltd. Both gel extraction kits and plasmids extraction kits were purchased from QIAGEN. The constructed plasmids were subsequently transformed into E. coli DH5α competent cells for further molecular cloning. The full sequences of the plasmid were provided in Supplementary Data 1.

**Protein purification**. NDM-1, apo-NDM-1, NDM-1-C208A were overexpressed and purified according to the procedures described previously[31]. Briefly, a single colony of E. coli BL21 (DE3) (TIANGEN Biotech(Beijing)Co., Ltd.) transformed with the plasmid encoding NDM-1 or its variant was inoculated into LB medium supplied with 50 µg·mL$^{-1}$ kanamycin and grown at 37 °C. Protein overexpression was induced using 0.2 mM isopropy-β-D-thiogalactoside (IPTG) supplemented with 0.2 mM ZnSO$_4$ at OD$_{600}$ 0.6. The bacterial culture was incubated at 25 °C overnight. To purify the respective protein, the cultured cells were harvested by centrifugation at 4500 × g and resuspended in a lysis buffer (20 mM HEPES, 0.5 M NaCl, and 1 mM PMSF at pH 7.0). The cells were ice-cooled and lysed by sonication and then centrifuged at 20,000 × g for 30 min to remove the majority of cell debris. The supernatant was filtered using Minisart syringe filter (0.45 µm) to remove any remaining large and insoluble cell debris, and was then applied to a 5 mL Ni(II)-loaded HiTrap chelating columns (GE Healthcare) at a rate of 2 mL·min$^{-1}$. The column was washed using five column volumes of washing buffer (20 mM HEPES, 0.5 M NaCl, and 30 mM imidazole at pH 7.0). The protein was eluted out using four column volumes of the same buffer with gradient. amounts of imidazole, and was subsequently dialyzed against the cleavage buffer (20 mM HEPES, 0.15 M NaCl at pH 7.0). The N-terminal His-tag of the fusion protein was cleaved by adding 50 NIH units of thrombin (Sigma-Aldrich, USA) at 25 °C for 3 h with mild shaking and the cleaved His-tag was separated from the resulting proteins by passing through the Ni(II)-NTA column again using washing buffer so that >90% of the proteins were in the flow-through fraction. The enzymes were further purified using HiLoad 16/60 Superdex 200 pg gel filtration column (GE Healthcare). The samples were then concentrated using Amicon Ultra-15 Centrifugal Filter Devices (Millipore) and separated into aliquots after dialysis with storage buffer (20 mM HEPES, 0.1 M NaCl at pH 7.0) for long-term storage at −80 °C. For the purification of MCR proteins, a single colony of E. coli BL21(DE3) transformed with the plasmid encoding MCR-1 or MCR-1-S was inoculated into LB medium supplied with 100 µg·mL$^{-1}$ ampicillin, and cultured at 37 °C overnight. Overnight cultures were 1:1000 amplified with fresh LB medium supplemented with 100 µg·mL$^{-1}$ ampicillin. Bacterial cultures were supplemented with 0.2 mM IPTG at OD$_{600}$ of 0.6, and subsequently, cultured at 25 °C for another 20 h. Cell pellets were collected by centrifugation at 4,500 × g for 30 min at 4 °C, and lysed by sonication in lysis buffer (20 mM HEPES, 50 mM NaCl, 20 mM imidazole at pH 7.4) at 4 °C and centrifuged to remove the majority of cell debris. The supernatant was collected by centrifuge at 18,000 × g at 4 °C for 45 min and filtered using Minisart syringe filter (0.45 µm) to remove remaining insoluble cell debris. The lysate was subjected to a 5 ml Ni(II)-loaded HiTrap chelating columns pre-washed by five column volumes of washing buffer (20 mM HEPES, 500 mM NaCl, and 30 mM imidazole at pH 7.4) at a rate of 2 mL·min$^{-1}$. The protein was eluted out using five column volumes of the washing buffer with gradient amounts of imidazole, and was subsequently incubated with 50 NIH units of thrombin for the following dialysis in cleavage buffer (20 mM HEPES,

10 mM NaCl at pH 7.4) at 4 °C overnight. The protein was reloaded onto another Ni(II)-NTA column to collect the flow-through fraction. Q-HP column (GE Healthcare) was employed for further purification, where gradient amounts of NaCl (10–500 mM) were used to elute protein sample. Target fractions were concentrated and loaded onto Superdex 75 (GE Healthcare) equilibrated with running buffer (20 mM HEPES, 50 mM $NH_4NO_3$, pH = 7.4). For the purification of full-length MCR-1 protein, an additional centrifugation at $80,000 \times g$ was performed to collect the insoluble membrane portion after the removal of cell debris. The resulting membrane protein was solubilized in PBS buffer supplemented with 2% (w/v) n-Dodecyl β-D-maltoside at 4 °C overnight. All other related buffers were similar to those for MCR-1-S purification but supplemented with 0.023% (w/v) DDM. For the preparation of apo-MCR-1-S, purified MCR-1-S was exposed to 50 molar equivalents of EDTA at 4 °C overnight and then concentrated to 1 mg·mL$^{-1}$. The protein was then extensively dialysis against a buffer (20 mM HEPES, pH 7.4, 50 mM $NH_4NO_3$) at 4 °C for 24 h.

**NDM-1 activity assay**. The NDM-1 activity assay was performed based on a previous reported method[34]. Briefly, NDM-1 or NDM-1-C208A (50 nM) were incubated with Au(PEt$_3$)Cl in enzyme assay buffer [50 mM HEPES buffer, 100 mM NaCl at pH 7.4) for 1 h at 25 °C, and then mixed with equal volume of 200 μM nitrocefin. The assay was performed in 96-well plate using the kinetic mode on a Varian Cary50 UV-visible spectrophotometer at 25 °C. The increase in absorbance at 490 nm was monitored every minute for a duration of 30 min until the reaction was completed. Linear portions of curves were used for data analysis. For the time-dependent incubation assay, NDM-1 (10 nM) were preincubated with Au(PEt$_3$)Cl (5 μM) for different time from 0 to 90 min, followed by the enzyme activity measurement using the method mentioned above.

**Michaelis-menten kinetics**. NDM-1 enzyme (50 nM) was incubated with Au (PEt$_3$)Cl at various concentrations (0, 0.5, 1, and 2 μM) in enzyme assay buffer for 1 h at 25 °C. Nitrocefin as a substrate was added to the enzyme to make the final substrate concentrations from 25 to 200 μM. Control experiment was also performed in the absence of inhibitors under the same conditions. The $K_m$, $V_{max}$, and $Ki$ for both the uninhibited and inhibited reactions were obtained by fitting the data into the double reciprocal Lineweaver–Burk plots.

**Zinc displacement analysis**. The purified NDM-1 (10 μM) or MCR-1-S (20 μM) was incubated with 50 μM $ZnSO_4$ or 100 μM zinc acetate (Zn(Ac)$_2$) in dialysis buffer [50 mM HEPES, 20 mM at pH 7.4) overnight at 4 °C, and the unbound Zn (II) ions were removed by dialysis in Zn-free dialysis buffer to ensure that Zn(II) was fully loaded into the proteins. The NDM-1 or MCR-1-S was then incubated with various concentrations of Au(PEt$_3$)Cl by dialysis at 4 °C overnight with mild shaking. The samples were subsequently dialyzed in dialysis buffer to remove unbound-metal ions, and were then acidified by concentrated $HNO_3$ at 60 °C for 4 h. Samples were diluted to a detectable concentration range and subjected to ICP-MS analysis (Agilent 7700x, Agilent Technologies, CA, USA) with $^{115}$In as an internal standard for $^{197}$Au, $^{66}$Zn. Protein concentrations were quantified by standard bicinchoninic acid (BCA) assay (Thermal Fisher Scientific, USA). The data were fitted by one-site binding Hill plot, and the maximal binding capacity (B$_{max}$) as well as the dissociation constant ($K_d$) were estimated.

**Zinc supplementation assay**. Au-NDM-1 or Au-MCR-1-S was prepared by incubation of apo-proteins with excess amounts of Au(PEt$_3$)Cl or AuCl overnight, followed by removal of unbound Au(I) and verification of the bound Au by ICP-MS. The above protein solutions were mixed with $ZnSO_4$ or with Zn(Ac)$_2$ at concentrations up to 50 molar equivalents for NDM-1 or 5 molar equivalents for MCR-1-S, and incubated for 4 h at 25 °C. Nitrocefin was added to the protein (50 nM) solutions and reaction rate was calculated as mentioned above.

**Zinc release assay**. Zinc release assay was carried out with a metallochromic indicator 4-(2-pyridylazo)resorcinol (PAR) based on a modified method as described previously[55]. Briefly, for PAR assay on NDM-1, Zn$_2$-NDM-1 (10 μM protein in 20 mM HEPES buffer, pH 7.4, 150 mM NaCl) was incubated with 20 molar equivalents of Au(PEt$_3$)Cl, for different time at 25 °C. The mixtures were then incubated with 150 μM 4-(2-pyridylazo) resorcinol (PAR) and UV spectra were then recorded. The absorbance at 490 nm is due to the formation of Zn(PAR)$_2$. Zn(II) contents were quantified in accordance with the standards. For PAR assay on MCR-1-S, all the procedures were similar except that MCR-1-S protein at 20 μM and Au(PEt$_3$)Cl at 10 molar equivalents were used.

**MALDI-TOF mass spectrometry**. The binding of AUR to NDM-1 was examined by matrix assisted laser ionization time of flight mass spectrometry (MALDI-TOF MS) (Bruker ultraflex extreme MALDI-TOF-TOF-MS). NDM-1 protein was incubated with 10 molar equivalents of AUR at 4 °C for 12 h. One microliter of the protein sample was mounted on a modified stainless-steel sample plate using electrically conductive tapes (9713 XYZ-Axis; 3 M, St. Paul, MN) and then overlaid with 1 μL of matrix solution (saturated sinapic acid in ACN: $H_2O$ = 50: 50). The plate was then introduced into the mass spectrometer, operating in the positive

reflectron mode. The mass resolution of ion peaks was recorded in the range of $m/z$ 20,000–40,000. MS data were processed using FlexAnalysis (version 1.2, Bruker Daltonics).

**X-ray crystallography**. Au-NDM-1 crystals were obtained by co-crystallization. Au-NDM-1 was prepared by incubating apo-NDM-1 with 10 eq. of AUR overnight at 25 °C and then remove the excess Au(I) by dialysis. Crystals of the Au-NDM-1 were grown by sitting-drop vapor diffusion method with the precipitant containing 100 mM MES (pH 5.5), 200 mM lithium sulfate, and 25% PEG 3350 (w/v). Tetragonal or diamond-like crystals appeared in 1 week and grew up to full size after two weeks. These crystals generally diffracted to the resolution ranging from 1.09 to 1.40 Å. The crystals were soaked in cryo-protectant solution [100 mM MES (pH 5.5), 200 mM lithium sulfate, 25% PEG 3350 (w/v), and 15% glycerol (w/v)] before frozen into liquid nitrogen. Au-MCR-1-S crystals were obtained by a soaking method according to a modified method as reported previous[31]. Crystal screening (sitting drop) was performed by mixing equal volume of reservoir and reservoir solution (100 mM KSCN, 30-32% PEG 3350, 100 mM Tris-HNO$_3$, pH 8.0). Crystals of native Zn-bound MCR-1-S appeared at 25 °C after two weeks. The crystals were transferred into wells supplemented with chelating solution (10 mM EDTA, 32% PEG 3350, 100 mM Tris-HNO$_3$, pH 8.0). Apo-MCR-1-S crystals were obtained after 12-h treatment, and washed three times with washing solution (32% PEG 3350, 100 mM Tri-HNO$_3$, pH 8.0), followed by soaking with 2 mM Au(PEt$_3$)Cl in soaking solution (32% PEG 3350, 25% glycerol, 100 mM Tri-HNO$_3$, pH 8.0) in darkness. The crystals were collected two weeks later and flash-frozen in liquid nitrogen.

All the diffraction data were collected at BL17U beamline of Shanghai Synchrotron Radiation Facility (SSRF) at the wavelengths of 0.979 Å. The presence of gold was confirmed by $mFo-DFc$ (difference Fourier) map with positive peaks (Au-NDM-1: ≥40σ, Au-MCR-1-S: ≥30σ) and anomalous peaks (Au-NDM-1: ≥5σ, Au-MCR-1-S: ≥18σ) corresponding to gold sites as well as X-ray excitation spectrum with an excitation peak for gold (Au-L3) at the radiation energy around 9.8 keV (Supplementary Fig. 6a and 8). The diffraction data were reduced with XDS[56]. The Phaser from CCP4 suite[57,58] and Phenix[59] were used for data refinement and finalization. Native NDM-1 structure (PDB ID: 5ZGE)[23] was used as a searching model for molecular replacement of Au-NDM-1 while the C-terminal catalytic domain of MCR-1 (PDB ID: 5GRR)[60] was used for that of Au-MCR-1-S. Cycles of refinement with the anomalous data and with careful manual rebuilding were done by using Refmac 5.8.0135[61] and Coot version 6[62], respectively. TLS refinement was used in the later stages of data processing. The Au occupancy was refined according to Au anomalous signal and was assessed based on atomic B-factor. The final models were analyzed with MolProbity 4.4[63]. Structural alignment was done over C$_α$ residues using DaliLite version 5. All of the structural illustrations were generated using the software PyMOL1.8.0.0. The coordinates and structure factors for Au-NDM-1, Au-MCR-1-S, Zn-MCR-1-S, and Apo-MCR-1-S were deposited at protein databank with accessing code 6HLE, 6LI6, 6LI4, and 6LI5, respectively. Details of the data collection and model refinement statistics are summarized in Supplementary Tables 1, 2, 4 and 5.

**Cellular thermal shift assay (CETSA)**. The cellular thermal shift assay was performed according to a general standard method[36]. Bacterial cultures (NDM-Rosetta or MCR-1-S-BL21) at logarithmic phase were exposed to AUR (12 μg·mL$^{-1}$ for NDM-Rosetta, 15 μg·mL$^{-1}$ for MCR-1-S-BL21) treatment overnight. The bacterial pellets were harvested and washed with PBS for three times. The cell suspensions were aliquoted into PCR tubes and heat treatment was performed at the designated temperature for 3 min in a 96-well thermal cycler. The tubes were cooled immediately at 25 °C, and the heat treatments were repeated for three cycles. For the cell lysis, the samples were frozen-thawed in liquid nitrogen and thermal cycler was set at 25 °C. The samples were vortexed gently after each cycle and centrifuged at $20,000 \times g$ to obtain the supernatant. All the samples were subjected to SDS-PAGE gel and electrotransferred to a PVDF membrane (Hybond-P, GE Healthcare). A PageRuler Prestained Protein Ladder #26616 (Thermo) was used as a standard marker. Diluted protein primary antibody [NDM-1 monoclonal antibody (NBP1-77688, NOVUS Biologicals)] or MCR-1 polyclonal antibody (CSB-PA745804LA01ENL, Cusabio Technology LLC) and the secondary antibody (Anti-rabbit IgG, HRP-linked Antibody, #7074, Cell Signaling Technology, Inc.) were applied after the standard blotting procedures. The protein bands were calorimetrically developed with specified ratio of substrates comprising nitroblue tetrazolium/5-bromo-4-chloro-3-indolyl phosphate (NBT/BCIP) for 15 min. Image J (1.52a)[64] was used to quantify the signals of each band for analysis. The software GraphPad Prism 8.0 (GraphPad Software, La Jolla CA, USA, www.graphpad.com) was used to analyze the resulting plots. Full scans of the blots were provided in Supplementary Data 2.

**Lysate activity assay**. Overnight cultures of NDM-Rosetta were diluted to OD$_{600}$ ~0.5 with fresh LB medium supplemented with AUR or Au(PEt$_3$)Cl at varying concentrations. The bacterial pellets were collected by centrifuge and washed with PBS for four times, followed by lysis via sonication in lysis buffer (20 mM HEPES, 50 mM NaCl, 2 mM TCEP at pH 7.4). The total protein concentrations of the lysate

were normalized to 1 mg·mL$^{-1}$ by BCA assay and subsequently subjected to activity assay as described in NDM-1 activity assay section.

**MCR-1 activity assay**. The enzymatic assay was performed according to a previous reported method with a slight modification[38]. Briefly, 0.1 mM full-length MCR-1 was treated with 10 molar equivalents of AuCl or Au(PEt$_3$)Cl at 25 °C for 18 h and subsequently incubated with 0.1 mM of a fluorescent substrate, nitrobenzodiazole-labeled glycerol-3-phosphoethanolamine (NBD-glycerol-3-PEA, Avanti Lipids, USA) in assay buffer (50 mM HEPES, pH 7.5, 100 mM NaAc, 0.023% DDM) at 25 °C overnight. TLC plate was used to separate NBD-glycerol from the MCR-1 reaction mixture in a mobile phase [ethyl acetate: methanol: water, 7:2:1 (vol/vol)]. The reaction product was analyzed by exposing the TLC plate to UV light (455–485 nm) and visualizing the fluorescent signals with a gel imaging system.

**Microdilution MIC assay**. Generally, bacteria were cultured in LB broth overnight at 37 °C, 250 rpm. The bacterial density was adjusted to about $1 \times 10^6$ CFU·mL$^{-1}$ and checked by CFU counting on agar plates afterwards. Tested antibiotics (MER or COL) or AUR or their combinations were added triplicately into individual wells of flat-bottomed 96-well plate and performed 2-fold serial dilution, followed by addition of prepared bacterial inocula. The plate was then incubated at 37 °C overnight. Wells with no drugs served as growth controls and wells with medium only served as background controls. The MIC was determined as the lowest concentration of a drug that could inhibit the growth of microorganism by both visual reading and OD$_{600}$ using a microtiter plate reader.

The FICI was determined according to the following equation: FICI = FIC$_A$ + FIC$_B$ = C$_A$/MIC$_A$ + C$_B$/MIC$_B$, where MIC$_A$ and MIC$_B$ are the MIC values of compounds A and B, respectively, when functioning alone, and C$_A$ and C$_B$ are the concentrations of compounds A and B at the effective combinations. Synergism was defined when FICI ≤ 0.5, indifference was defined when FICI > 0.5 and < 4, and antagonism was defined when FICI ≥ 4. All of the determinations were performed at least in triplicate on different days.

**Time-kill assay**. In a typical assay, bacteria (NDM-HK or MCR-1-J53) were cultured overnight and diluted 1:100 into LB broth at 37 °C for 3 h to reach log phase. The initial bacterial density was adjusted to ~10$^5$–10$^6$ CFU·mL$^{-1}$ and then exposed to antibiotic, gold compounds either alone or in combination. LB broth with no drugs served as a control. Aliquots of bacterial suspension were withdrawn at different time intervals within 24 h for inspection of bacterial viability by agar plating. The concentrations of the drugs used were 16 μg·mL$^{-1}$ MER and 30 μg·mL$^{-1}$ AUR for NDM-1 assay and 2 μg·mL$^{-1}$ COL and 6 μg·mL$^{-1}$AUR for MCR-1 assay. Data from three independent experiments were averaged and plotted as log$_{10}$ CFU·mL$^{-1}$ versus time (h) for each time point over 24 h. All assays were triplicated and performed three times on different days.

**Membrane potential assay**. Membrane potential assay was carried out based on a modified method as previously described[40] and the manufacturer's instructions of BacLight bacterial membrane potential kit (B34950, Thermo fisher). Briefly, about 10$^6$ CFU mid-log phase bacterial pellets (*S. flexneri* (MCR-1$^+$) or *S. flexneri* (MCR-1$^-$)) that were preincubated with AUR (3 μM), Au(PEt$_3$)Cl (3 μM) or CCCP (5 μM) at fixed concentration, were collected, and washed by PBS for three times. The bacterial pellets were then resuspended in PBS supplemented with DiOC2 (30 μM) and stained at 37 °C for 30 min. Stained bacteria were then assayed in a flow cytometer and the signals from FITC-A (488 nm, green gate) and PI-A (633 nm, red gate) were collected and then analyzed by FlowJo v10 (Tree Star Inc., Ashland, USA). *S. flexneri* without treatment served as the control. The assays were performed in triplicate, repeated three times and results were expressed with fluorescence ratio (green/red) as average ±SD.

**Resistance study**. Briefly, bacteria (NDM-HK or MCR-1-J53) at ~$1 \times 10^{10}$ CFU was plated onto LB agar containing antibiotics (MER or COL) and AUR at different concentrations, and incubated at 37 °C. After incubation for 48 h, to any plates with observable colonies, up to four colonies were picked and recultured, followed by the measurement of their MIC values. Any MIC that was greater than the original value was determined as higher-level resistant mutant colony. The concentration that restricted the growth of mutant colonies was determined as MPC. At the identical experiment, the higher-level resistant mutant colonies were enumerated. The relative mutation frequency for each MIC for each strain/antibiotic pair was calculated as the proportion of resistant colonies per inoculum. The serial passage assay was performed by a method described previously[31]. Briefly, an overnight culture of bacteria (NDM-HK or MCR-1-J53) was diluted to ~10$^7$ CFU·mL$^{-1}$ in LB broth. The as-diluted bacterial suspension was added to each well of 96-well plate supplemented with drugs at escalating concentrations. All the plates were incubated at 37 °C and the growth of cultures was checked at 24-h intervals. Cultures from the second highest concentrations that allowed growth were performed 1:1000 dilution into fresh medium supplemented with the same concentrations of drugs. For MER, 1 fold of MIC was set as 16 μg·mL$^{-1}$. For the combination of MER and AUR, 1 fold of MIC was set as 2 μg·mL$^{-1}$ MER + 8 μg·mL$^{-1}$ AUR. For COL, 1 fold of MIC was set as 8 μg·mL$^{-1}$. For the combination of COL and AUR, fold of MIC was set as 2 μg·mL$^{-1}$ COL + 5 μg·mL$^{-1}$

AUR. This in vitro passage was repeated for 16 days. MIC of MER was determined every two passages.

**Ethics statement**. The ethics Committee on the Use of Live Animals in Teaching & Research (CULATR) of the University of Hong Kong (HKU) approved all of the protocols (Reference code: 4008-16 and 5079-19) and procedures in this study.

*Animals*: Six to eight weeks old, female, 18–22 g of weight Balb/c mice were purchased from Charles River Laboratories, Inc. and used in all mouse studies. The animals were numbered and allocated into groups using a simple randomization of excel-generated random numbers. To avoid biases, we also assured that different treatments were performed on the same day. All the animals were randomized to cages for each experiment and had free access to food and water.

**Murine peritonitis infection**. The murine infection experiments were modified according to previously described methods[8]. For the bacterial load experiment, an overnight culture of *K. pneumoniae* 9607 (MCR-1$^+$) was performed 1:250 dilution in LB broth and regrew to about OD$_{600}$ 0.5 at 37 °C, 250 rpm. Bacterial pellets were collected and washed by PBS buffer three times for further use. Mice were infected intraperitoneally (*i.p.*) with a dose of $2 \times 10^6$ CFU of bacteria in PBS. Four groups of mice were *i.p.* administered 1-h post-infection with single dose of vehicle, COL (2 mg·kg$^{-1}$), AUR (0.5 mg·kg$^{-1}$), or their combination therapy (*n* = 4). All the mice were scarified at 48 h following systemic infection, and bacterial loads in livers and spleens were examined by agar plating. For survival assay, all the operations of infection were similar, except that infection was induced by *E. coli* CKE at $5 \times 10^6$ CFU per mice in the presence of 2% mucin, and single dose of vehicle, AUR (0.25 mg·kg$^{-1}$), COL (1 mg·kg$^{-1}$) and their combination was *i.p.* injected to the infected mice 0.5-h post-infection (*n* = 6). Body weights and mice survival were monitored till endpoint of the experiment.

**Statistical analysis**. All statistical analyses were performed on three independent experiments, or more if otherwise stated, using Prism 8.0 (GraphPad Software Inc.) software. Sample size and information about statistical tests are reported in the figure legends and Methods. Data are presented as mean ± SEM.

**Reporting summary**. Further information on research design is available in the Nature Research Reporting Summary linked to this article.

## Data availability

The protein crystal structure data for Au-NDM-1, Zn-NDM-1, Au-MCR-1-S, Zn-MCR-1-S, and apo-MCR-1-S have been deposited at Protein Data Bank (PDB) with accessing codes of 6LHE, 5ZGE, 6LI6, 6LI4, and 6LI5, respectively. All other data supporting the findings of this study are available from the corresponding author (Hongzhe Sun: hsun@hku.hk) on request. Source data are provided with this paper.

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

## Acknowledgements

We thank Drs. Menglong Hu, Yi Wang, and Ms. Yuan Wu for helpful discussion on crystallographic study; Dr. Ya Wang for help in microbiology experiments; Dr. Kwan-Ming Ng for help in MALDI-TOF-MS, and Dr. Peng Gao for help in animal study. NDM-1-positive *K. pneumonia* isolate was kindly given by Prof. Patrick-Chiu Yat Woo. The crystal diffraction data were collected at Shanghai Synchrotron Radiation Facility (SSRF), the Chinese Academy of Sciences. We thank the staff at BL17U1 beamline of SSRF for their generous help. This work was supported by Research Grants Council of Hong Kong (R7070-18, 17307017), Health and Medical Research Fund (HKM-15-M10) and Seed Fund for Basic Research (201910159244).

## Author contributions

H.S., H.L., and P.-L.H. conceived idea and designed experiments; R.W., Q.Z., and Y.-T. W. constructed the plasmids, overexpressed, purified the proteins, and performed all enzyme-based and cell-based experiments; Q.Z., H.W., and M.W. crystallized the

proteins and solved the structures by X-ray crystallography; R.W. and Q.Z. performed animal experiments. Q.H. and A.Y. assisted in analysis of crystallography data. P.-L.H. and R.Y.-T.K. aided the biological interpretation of the microbiological and animal data. H.L., R.W., and H.S. drafted and revised the manuscript with input from all co-authors. All authors approved the final version of the manuscript.

## Competing interests

H.S., P.-L.H., Q.Z., R.W., and H.L. have filed a patent application related to the work of this manuscript. The rest of the authors declare no competing interests.
