## [Peer Review File · Nature Communications]

REVIEWER COMMENTS

Reviewer #1 (Remarks to the Author):

Antibiotic resistance in Gram-negative bacteria is spreading worldwide and last resort antibiotics such as carbapenems are becoming less effective. This is due mostly to the action of carbapenemases, such as the Metalloenzyme NDM-1. Based on this, colistin (an antibiotic with toxic secondary effects) has been recently reused in the clinics. However, colistin resistance has been reported due to the presence of the Zn enzyme MCR-1. Several strains have been reported carrying genes coding for both metalloenzymes.

In this work, Wang and coworkers report the use of auranofin (AUR), a gold-based anti-rheumatic drug, as an inhibitor of both NDM-1 and MCR. They claim that the action of AUR is due to the displacement of the Zn(II) cofactors from the active sites in both enzymes, and they demonstrate that AUR potentiates the action of antibiotics on resistant bacteria. Successful experiments on mice as infection models also support the potency of this drug.

This work is outstanding, in the sense that targets at the same time two Zn(II)-dependent enzymes that are completely unrelated in terms of structure, mechanism and active site. There are some aspects that the authors may want to consider to improve the quality of their work:

1. They conclude that inhibition is through a competitive mechanism. This is concluded from only two experiments at each concentration of inhibitor in the double reciprocal plot. More data are required to support this assertion.
2. The inhibition of the Cys208Ala mutant of NDM-1 (use of the consensus BBL numbering is advised for MBLs) by AUR is less potent compared to wild type NDM-1, and they attribute this to the presence of the sulfur atom in AUR and in the Cys ligand. However, the proposed model of inhibition relies in the finding that the Au atoms replace the Zn(II). This does not require the involvement of the thiol moiety. Moreover, formation of a disulfide bridge between AUR and Cys221 (BBL numbering) in wild type NDM-1 would thwart binding of Au to the Zn center.
3. Mutation of the Cys ligand in MBLs abrogates the lactamase activity in many cases. Which is the resistance conferred by this mutant in bacteria?
4. The crystal structure of the Au-derivative of NDM-1 was obtained by addition of Au to apo-NDM-1, in contrast to the mechanism of inhibition by Bi(III) shown by this group. Are there differences between these two metal ions? Measurement of the affinity constants could be of help to identify differences, unless the authors show that there is a kinetic barrier. However, in this case, this should be driven by the koff values of the Zn(II) ions.
5. For the titration with the chelating agent PAR, the authors should present a control experiment without Au.

Reviewer #2 (Remarks to the Author):

This is a very interesting study showing that the FDA approved drug auranofin inhibits both Zn-metallo-beta-lactamases and the polymyxin resistance enzyme MCR-1. The authors provide a comprehensive study of inhibition in vitro including X-ray structures of both NDM and MCR showing Au has replaced Zn in the active sites. The inhibition of MCR is surprising in that it does not have a cysteine residue in the active site. In addition, the authors show that auranofin acts synergistically with carbapenems and colistin against Gram-negative bacteria expressing metallo-beta-lactamases as well as MCR. Finally, the authors use a mouse infection model to show auranofin and colistin shows potency in vivo. Taken together, these studies are supportive of potential of auranofin as an adjuvant to circumvent resistance against carbapenems and polymyxins. Specific points to address are listed below.

1. I understand that this is an approved drug, but nevertheless, it would be of interest to know the specificity of AUR versus other Zn metallo-enzymes such as carbonic anhydrase. If this is known the authors should discuss it. If not, test AUR against carbonic anhydrase or another model Zn enzyme.
2. It would also be useful to include some discussion of why enzyme catalysis is inhibited by the replacement of Zn by Au. They look to occupy similar positions in the structure. Au is a soft metal.
3. Page 2, line 74. It is not apparent from Supplemental Fig 1B, C that nickel inhibits bacterial growth. In Fig 1B, Ni looks to have no effect and in 1C I don't see the Ni symbols on the figure.
4. Page 2, line 52. Note that Stojanoski et al (BMC Biol. 14:81) were the first to publish the MCR-1 structure and to show that Thr285 is required for catalysis.
5. Page 2, lines 87-89, Fig. 1a. The data clearly show inhibition after 1hr incubation. It would be of interest to show if there is a time-dependence to the inhibition. One might expect time-dependence since Au is replacing Zn.
6. Page 4, lines 163-187. Figure 3 is missing.

7. Page 5, line 227. I suggest rather than “against a variety of E. coli strains producing”, state “against an E. coli strain produced different MCR variants”, since they are all E. coli J53.

TEL: (852) 2859 8974
FAX: (852) 2857 1586
E-mail: hsun@hku.hk

DEPARTMENT OF CHEMISTRY
THE UNIVERSITY OF HONG KONG
POKFULAM ROAD, HONG KONG

Prof. Hongzhe Sun

Re: “Resensitizing superbugs carrying both *bla*_{MBL} and *mcr* genes to antibiotics by auranofin”
(your previous manuscript no: NCOMMS-20-15301)

We appreciate both reviewers’ favorable comments and helpful suggestions!

Reviewer #1 (Remarks to the Author):

Antibiotic resistance in Gram-negative bacteria is spreading worldwide and last resort antibiotics such as carbapenems are becoming less effective. This is due mostly to the action of carbapenemases, such as the Metalloenzyme NDM-1. Based on this, colistin (an antibiotic with toxic secondary effects) has been recently reused in the clinics. However, colistin resistance has been reported due to the presence of the Zn enzyme MCR-1. Several strains have been reported carrying genes coding for both metalloenzymes. In this work, Wang and coworkers report the use of auranofin (AUR), a gold-based anti-rheumatic drug, as an inhibitor of both NDM-1 and MCR. They claim that the action of AUR is due to the displacement of the Zn(II) cofactors from the active sites in both enzymes, and they demonstrate that AUR potentiates the action of antibiotics on resistant bacteria. Successful experiments on mice as infection models also support the potency of this drug. This work is outstanding, in the sense that targets at the same time two Zn(II)-dependent enzymes that are completely unrelated in terms of structure, mechanism and active site. There are some aspects that the authors may want to consider to improve the quality of their work:

1. They conclude that inhibition is through a competitive mechanism. This is concluded from only two experiments at each concentration of inhibitor in the double reciprocal plot. More data are required to support this assertion.

Response: We truly appreciate the reviewer’s comments and suggestions. The inhibition kinetics assay was re-performed as suggested (**Fig. 1**). We confirmed AUR (as Au(PEt₃)Cl in this assay) inhibits the activity of NDM-1 through a non-competitive mode, in consistence with what was stated in original manuscript. The kinetics were re-calculated based on the newly-collected data. We’ve added the data as **Fig. 1b** in the revised version and the related description has been reworded in the revised manuscript (Page 2, Line 88-92).

TEL: (852) 2859 8974
 FAX: (852) 2857 1586
 E-mail: hsun@hku.hk

DEPARTMENT OF CHEMISTRY
 THE UNIVERSITY OF HONG KONG
 POKFULAM ROAD, HONG KONG

Prof. Hongzhe Sun

Figure 1 Double reciprocal plot of substrate dependent enzyme kinetics on inhibition of NDM-1 activity by Au(PEt₃)Cl.

2. The inhibition of the Cys208Ala mutant of NDM-1 (use of the consensus BBL numbering is advised for MBLs) by AUR is less potent compared to wild type NDM-1, and they attribute this to the presence of the sulfur atom in AUR and in the Cys ligand. However, the proposed model of inhibition relies in the finding that the Au atoms replace the Zn(II). This does not require the involvement of the thiol moiety. Moreover, formation of a disulfide bridge between AUR and Cys221 (BBL numbering) in wild type NDM-1 would thwart binding of Au to the Zn center.

Response: This reviewer's helpful comments are highly appreciated. Through the mutagenesis study, we concluded the binding of Cys208 (Cys221 in BBL numbering) with AUR is important for the following inhibitory action of AUR. And the binding of AUR to NDM-1 is via its active form of [Au(PEt₃)]⁺ or Au⁺ as suggested by MALDI-TOF-MS experiment (Fig. S5). We agree that it is premature to assert that this process required the involvement of the thiol moiety, since Au(PEt₃)Cl was shown to suppress NDM-1 activity (Fig. 1a) and displace Zn(II) with Au(I) (Fig. 1d). To further clarify this issue, the related description has been reworded to "..., indicating that the interaction of AUR with Cys208 is crucial for its inhibition on NDM-1." in the revised manuscript (Page 2, Line 97).

3. Mutation of the Cys ligand in MBLs abrogates the lactamase activity in many cases. Which is the resistance conferred by this mutant in bacteria?

Response: The mutagenesis performed at position Cys208 (Cys221 in BBL numbering), would render most of the resulting C208A mutant as a mono-Zn(II) (Zn1-bound) enzyme. We repeated the enzyme assay and still observed ~15% residual activity remained for the C208A mutant in

TEL: (852) 2859 8974
 FAX: (852) 2857 1586
 E-mail: hsun@hku.hk

DEPARTMENT OF CHEMISTRY
 THE UNIVERSITY OF HONG KONG
 POKFULAM ROAD, HONG KONG

Prof. Hongzhe Sun

comparison to wild type NDM-1 as shown in Fig. 2. The same phenomenon was also reported for the susceptibility test *E. coli* expressing IMP-1-C221A variant previously (Horton L. B. *et al*, *Antimicrob. Agents Chemother.* 2012, **56**, 5667-5677). The MIC values of β -lactam substrates are listed in Table 1. It has been demonstrated that B1 MBL could be active in mono-Zn(II) form, e.g. mononuclear MBL, mono-Zn(II) BcII (Vila J. *et al*, *Acc. Chem. Res.* 2006, **39**, 721-728) and mono-Zn(II) NDM-15 (Cheng Z, Crowder MW *et al*, *J. Biol. Chem.* 2018, **293**, 12606-12618), despite the hydrolytic activity of mono-Zn(II) NDM-1 was rarely reported. It is generally accepted that the carbonyl oxygen of the β -lactam ring coordinates to Zn1 during substrate binding, and the shared hydroxide will act as a nucleophile to attack the carbonyl carbon. The debate resides in whether the alanine substitution at 221 (BBL numbering) affords sufficient space for a water molecule or hydroxy ligand between the Ala methyl group and Zn2, and to what degree this would hinder the binding of substrate. Although resistance conferred by this mutant in bacteria may need further investigation (e.g. by crystallography study) in future, this mutagenesis study still concretely shows that the Cys208 is crucial for AUR inhibition on NDM-1.

Figure 2 Inhibition of NDM-1-C208A by Au(PEt₃)Cl. Note the low activity of the enzyme and limited inhibition by Au(PEt₃)Cl. Mean values of three replicates are shown and error bars indicate \pm SEM.

Table 1* Antibiotic susceptibilities of *E. coli* expressing position 221 IMP-1 variant enzymes (representative mutants are shown)

Antibiotic substrate	MIC (μ g/mL)							
	E. coli XL1-Blue	Wild type	C221A	C221D	C221G	C221S	C221T	C221V
Ampicillin	2	125	2	128	128	3	2	2
Cefotaxime	0.094	250	0.19	0.38	3	0.19	0.064	0.047

TEL: (852) 2859 8974
 FAX: (852) 2857 1586
 E-mail: hsun@hku.hk

DEPARTMENT OF CHEMISTRY
 THE UNIVERSITY OF HONG KONG
 POKFULAM ROAD, HONG KONG

Prof. Hongzhe Sun

Ceftazidime	0.38	500	0.38	48	64	0.38	0.19	0.19
Imipenem	0.25	2	0.38	0.5	1	1	1	0.25
Meropenem	0.032	1.5	0.094	1	0.38	0.094	0.094	0.064
Aztreonam	0.125	0.094	0.094	0.064	0.094	0.125	0.094	0.094

*The data are quoted from other's report (Horton, L. B. et al. *Antimicrob. Agents Chemother.* 2012, **56**, 5667-5677)

4. The crystal structure of the Au-derivative of NDM-1 was obtained by addition of Au to apo-NDM-1, in contrast to the mechanism of inhibition by Bi(III) shown by this group. Are there differences between these two metal ions? Measurement of the affinity constants could be of help to identify differences, unless the authors show that there is a kinetic barrier. However, in this case, this should be driven by the koff values of the Zn(II) ions.

Response: The protein crystallization is affected by many factors e.g. pH, temperature, ionic strength in the crystallization solution, and the affinity between drug and protein is definitely one of the key factors. Based on the data from our previous research (Fig. 2d, Wang R, Ho PL, Li H, Sun H *et al.*, *Nat. Commun.* 2018, **9**, 439), the apparent dissociation constant (K_d) of Bi(III) (as colloidal bismuth subcitrate) to NDM-1 was estimated to be 13.24 μM ($B_{max} = 1.002$). Under similar conditions, the K_d value of Au(I) (as Au(PEt₃)Cl) to NDM-1 was 4.03 μM ($B_{max} = 1.95$) (Fig. 1d in the original manuscript), which was ~one third of that of Bi(III). Therefore, the lower kinetic barrier between Au(I) and NDM-1 protein may increase the homogeneity of Au-liganded NDM-1 in crystallization reservoir solution, thus favoring the direct crystallization of Au-NDM-1.

5. For the titration with the chelating agent PAR, the authors should present a control experiment without Au.

Response: We thank this reviewer's careful reading. The data labelled with "Time 0" presents the control experiment without Au(I). Supplemental Fig. 3a were revised to clarify the confusion.

Reviewer #2 (Remarks to the Author):

This is a very interesting study showing that the FDA approved drug auranofin inhibits both Zn-metallo-beta-lactamases and the polymyxin resistance enzyme MCR-1. The authors provide a comprehensive study of inhibition in vitro including X-ray structures of both NDM

TEL: (852) 2859 8974
FAX: (852) 2857 1586
E-mail: hsun@hku.hk

DEPARTMENT OF CHEMISTRY
THE UNIVERSITY OF HONG KONG
POKFULAM ROAD, HONG KONG

Prof. Hongzhe Sun

and MCR showing Au has replaced Zn in the active sites. The inhibition of MCR is surprising in that it does not have a cysteine residue in the active site. In addition, the authors show that auranofin acts synergistically with carbapenems and colistin against Gram-negative bacteria expressing metallo-beta-lactamases as well as MCR. Finally, the authors use a mouse infection model to show auranofin and colistin shows potency in vivo. Taken together, these studies are supportive of potential of auranofin as an adjuvant to circumvent resistance against carbapenems and polymyxins. Specific points to address are listed below.

1. I understand that this is an approved drug, but nevertheless, it would be of interest to know the specificity of AUR versus other Zn metallo-enzymes such as carbonic anhydrase. If this is known the authors should discuss it. If not, test AUR against carbonic anhydrase or another model Zn enzyme.

Response: We thank this reviewer's helpful comments and suggestions. The specificity of Au(I)-based compounds against other zinc-enzymes is important for the following application in animals and even humans. Previous reports have indicated that AUR could disrupt the function of Zn(II) proteins, *e.g.* Trx2 (Harbut MB *et al*, *PNAS*, 2015, **112**, 4453–4458), Poly(ADP-ribose) polymerase (Mendes F, Casini A *et al*, *J. Med. Chem*, 2011, **54**, 2196–2206) and HIV-NCp7 (De Paulade Q, Farrell N *et al*, *J. Inorg. Biochem.*, 2009, **103**, 1347–1354). But whether AUR and related Au(I)-based compounds is active against other Zn(II) proteins/enzymes such as carbonic anhydrase may need further specific investigation.

2. It would also be useful to include some discussion of why enzyme catalysis is inhibited by the replacement of Zn by Au. They look to occupy similar positions in the structure. Au is a soft metal.

Response: This reviewer's helpful comments are highly appreciated.

In general, Zn(II) possesses a filled d^{10} orbital, and is not involved in redox reactions but rather serves as a Lewis acid. In addition, Zn(II) shows a ligand-field stabilization energy of zero in all liganding geometries (Huheey JE *et al*. 1993, *Inorganic Chemistry: Principles of Structure and Reactivity* (4th ed.), vol. 1 Harper Collins College Publishers New York.), and hence no geometry is inherently more stable than another. The negligible energetic barrier to a multiplicity of equally accessible coordination geometries could be adopted by Zn(II) enzymes to modulate the its activity, which might be an key factor for Zn(II)-dependent catalysis through changing in the Zn(II) coordination geometry in active site. Therefore, Zn(II) could serve as an ideal cofactor for reactions that require a Lewis acid-type catalyst (Butler A., *Science*, 1998, **281**, 207–209).

Specifically, Zn(II) has been reported to be critical for the function of both NDM-1 and MCR-1 (Feng H *et al.*, *Nat. Commun.* 2017, **8**, 2242; Hu M, Hao Q *et al*, *Sci. Rep.* 2016, **6**, 38793). In the

TEL: (852) 2859 8974
FAX: (852) 2857 1586
E-mail: hsun@hku.hk

DEPARTMENT OF CHEMISTRY
THE UNIVERSITY OF HONG KONG
POKFULAM ROAD, HONG KONG

Prof. Hongzhe Sun

intact NDM-1, Zn1 both Zn(II) ions adopt tetrahedral coordinate geometry with a [(His)₃Zn(μ-OH)] and [(His)(Asp)(Cys)Zn(μ-OH)] motif for Zn1 and Zn2 respectively. The two Zn(II) ions are bridged by a hydroxide ion with a zinc-to-zinc distance of ~4.6 Å. It is presumed that the Zn(II) plays an essential role to stabilize the formed intermediate during the attack of carbonyl carbon atom of the β-lactam ring by an activated hydroxide. Moreover, the water molecule should be activated by ionization, polarization, or poised for displacement once within the Zn(II) coordination sphere. In contrast, the activation of water molecule would barely be accomplished by Au(I). Furthermore, the metal-metal distance in Au-liganded NDM- is significantly shortened (~3.8 Å) in comparison to Zn-Zn distance (~4.6 Å) in native NDM-1. Two Au(I) ions are coordinated to two respective hydroxide ligands, and no well-ordered oriented water molecule is present between the metal ions, and the Asp124 carboxyl group coordinates instead. Under such rearrangement, the active site may apparently be in a conformation that is not optimal for the binding and hydrolysis of the substrates. Similarly, for the intact MCR-1, Zn(II) must undergo a change in coordination in active site and therefore could stabilize the formation of transition-state intermediate, *i.e.*, TPO285/lipid A. (Stojanoski V et al, *BMC Biol.*, 2016, **14**, 81). This process could also be disrupted by the displacement of Au(I).

Au(I), a soft metal, occupies similar positions with Zn(II) in the structure. However, the preferred coordination number and geometry of Au(I) is different from that of Zn(II) (Barber-Zucker S *et al*, *Sci. Rep.* 2017, **7**, 16381; Carugo O, *J. Inorg. Biochem.* 2017, **175**, 244–247). For example, Au(I) prefers two-coordination with four-coordination being less frequent, and hardly has five-coordination. Moreover, ligand exchange in Au(I) complex is much slower than that of Zn(II), particularly for tetrahedral Au(I) complexes (Pacheco EA, Tiekink ERT, Whitehouse MW in *Gold Chemistry: applications and future directions in the life sciences*. Ed. Mohr F, Wiley-VCH, Weinheim, 2009). These differences and affinities towards ligands/water between Zn(II) and Au(I) may largely alter the activated energy of transition state and thus makes the catalytic process no longer work under the same condition, leading to enzyme inhibition. A brief discussion on the effect of replacement of Zn(II) by Au(I) is included in the revised version.

3. Page 2, line 74. It is not apparent from Supplemental Fig 1B, C that nickel inhibits bacterial growth. In Fig 1B, Ni looks to have no effect and in 1C I don't see the Ni symbols on the figure.

Response: We thank this reviewer's careful reading and apologize that this was a typo. The active metal ions should be cobalt(II) instead of nickel(II). The colors and shape of the labels in the Supplemental Fig. 1b and 1c have been changed to make it easier to read. The related descriptions have been corrected accordingly in the revised manuscript (Page 2, Line 73).

TEL: (852) 2859 8974
FAX: (852) 2857 1586
E-mail: hsun@hku.hk

DEPARTMENT OF CHEMISTRY
THE UNIVERSITY OF HONG KONG
POKFULAM ROAD, HONG KONG

Prof. Hongzhe Sun

4. Page 2, line 52. Note that Stojanoski et al (BMC Biol. 14:81) were the first to publish the MCR-1 structure and to show that Thr285 is required for catalysis.

Response: We thank this reviewer for helpful comments. We've cited the reference in the revised manuscript as suggested (ref 28, Page 2, Line 53).

5. Page 2, lines 87-89, Fig. 1a. The data clearly show inhibition after 1hr incubation. It would be of interest to show if there is a time-dependence to the inhibition. One might expect time-dependence since Au is replacing Zn.

Response: In a pre-lab experiment, we've measured the activity of NDM-1 proteins in the presence of Au(PEt₃)Cl for different times. As shown in **Figure 3**, the NDM-1 activity was gradually decreased as the preincubation time increased, and was completely lost after around 1-hr incubation with Au(PEt₃)Cl. We therefore selected 1 hour as the preincubation time for the enzyme inhibition assay.

Figure 3 Inhibition of activity of NDM-1 that was preincubated with Au(PEt₃)Cl for different time.

6. Page 4, lines 163-187. Figure 3 is missing.

Response: We thank this reviewer's careful reading. The Fig. 3 has been added in the revised manuscript.

TEL: (852) 2859 8974
FAX: (852) 2857 1586
E-mail: hsun@hku.hk

DEPARTMENT OF CHEMISTRY
THE UNIVERSITY OF HONG KONG
POKFULAM ROAD, HONG KONG

Prof. Hongzhe Sun

7. Page 5, line 227. I suggest rather than “against a variety of E. coli strains producing”, state “against an E. coli strain produced different MCR variants”, since they are all E. coli J53.

Response: We thank this reviewer’s helpful comments and suggestions. The sentence has been reworded as suggested in the revised manuscript (Page 5, Line 230).

REVIEWERS' COMMENTS

Reviewer #1 (Remarks to the Author):

The authors have done a great job in addressing all my concerns and questions. This has resulted in a great final manuscript, which represents an outstanding contribution to the field.

Reviewer #2 (Remarks to the Author):

The authors have responded appropriately to the points in my initial review and have modified the manuscript to add clarity.

With regard to point 5, the response figure answers the question of time dependence. This should be included in supplemental figures and brief mention in the manuscript.

The response to point one with regard to disrupting the function of other Zn(II) enzymes is of interest but has not been included in the manuscript. The authors should consider adding a sentence about action on other Zn enzymes and potential toxicity.

TEL: (852) 2859 8974
FAX: (852) 2857 1586
E-mail: hsun@hku.hk

DEPARTMENT OF CHEMISTRY
THE UNIVERSITY OF HONG KONG
POKFULAM ROAD, HONG KONG

Prof. Hongzhe Sun

Re: Manuscript number: NCOMMS-20-15301B-Z

“Resensitizing carbapenem- and colistin-resistant bacteria to antibiotics using auranofin”

We appreciate both reviewers’ favorable comments and helpful suggestions!

Reviewer #1 (Remarks to the Author):

The authors have done a great job in addressing all my concerns and questions. This has resulted in a great final manuscript, which represents an outstanding contribution to the field.

Response: We thank this reviewer’s kind comments.

Reviewer #2 (Remarks to the Author):

The authors have responded appropriately to the points in my initial review and have modified the manuscript to add clarity.

With regard to point 5, the response figure answers the question of time dependence. This should be included in supplemental figures and brief mention in the manuscript.

The response to point one with regard to disrupting the function of other Zn(II) enzymes is of interest but has not been included in the manuscript. The authors should consider adding a sentence about action on other Zn enzymes and potential toxicity.

Response: We thank this reviewer’s helpful suggestions.

(1) The data of the time-dependent NDM-1 activity assay has been incorporated as Supplementary Figure 3c and the related description sentence has added in the main manuscript in the revised version (Line 105-107).

(2) A sentence has been added to describe the action on other Zn enzymes and potential toxicity of auranofin and Au(I) based compound has been mentioned in the revised manuscript (L325-326)